# The oldest sepioid cephalopod from the Cretaceous discovered by Digital fossil-mining with zero-shot learning AI
Kanta Sugiura [1,9], Shin Ikegami [1,9], Yusuke Takeda [2], Jörg Mutterlose [3], Mehmet Oguz Derin [4], Aya Kubota [5], Harufumi Nishida[6], Kazuki Tainaka [7], Takahiro Harada[4], Neil H. Landman [8] & Yasuhiro Iba [1] ✉

Sepioids are an evolutionarily successful group of modern ten-armed cephalopods (Decabrachia) of high biodiversity, providing a large amount of biomass in present-day oceans. They include the internally shelled order Sepiida (cuttlefish) and the soft-bodied order Sepiolida (bobtail squid). The phylogenetic position and evolutionary history of these orders are, however, so far poorly understood due to the patchy fossil record of the Decabrachia. Here we report *Uluciala rotundata* gen. et sp. nov. from the upper Campanian to upper Maastrichtian (~74–67 Ma, Upper Cretaceous), South Dakota, which shows an intermediate morphology between Sepiida and Sepiolida. This discovery was facilitated by a new approach in palaeontology, the Digital fossil-mining method incorporating a zero-shot learning AI model. *Uluciala rotundata* demonstrates a close relationship between the two sepioid orders, which has previously been interpreted controversially. Our findings indicate that sepioids experienced an early phase of radiation in the later part of the Late Cretaceous.

Cephalopods have evolved for 500 million years, being highly abundant and diverse in present and past oceans[1,2]. Fossil cephalopod assemblages from the Palaeozoic to Mesozoic were dominated by externally shelled forms, such as ammonoids and nautiloids[3]. In contrast, extant cephalopods are internally shelled or soft-bodied coleoids, with the exception of modern nautiloids.

Sepioids are a group of modern ten-armed coleoids (superorder Decabrachia) that includes two taxonomic orders, Sepiida (cuttlefish) and Sepiolida (bobtail squid) (Fig. 1)[4]. The Sepiida possess a mineralized internal shell (phragmocone), and the Sepiolida have a demineralized gladius or are entirely missing such a supporting structure[5]. Modern sepioids thrive in the shallow areas of all oceans, contributing substantially to the current marine biomass[6]. In terms of biodiversity, about 20% of the extant cephalopod species are represented by sepioids[7]. They are one of the most common cephalopod groups in coastal areas[8].

The early evolutionary history and the phylogenetic relationship of the two sepioid orders, however, remain poorly understood. The Sepiida provided abundant fossil records from Cenozoic sediments, but their early evolution in the Mesozoic is documented by only one specimen from the latest Cretaceous (*Ceratisepia vanknippenbergi*; ~70 Ma)[9,10]. The Sepiolida on the other hand have no reliable fossil record[11,12] because of their poor fossilization potential. This is explained by the absence of a mineralized shell and a high ammonia content in the soft tissues[13]. This extreme rarity of fossil findings limits the understanding of the early evolution of sepioids. Morphological and molecular analysis of modern taxa do not help to solve this problem without fossil evidence[14]. The beaks of sepioids can help to overcome these limitations due to their high fossilization potential, exceeded only by that of mineralized shells[15]. The beaks are unique hard tissues of cephalopods, consisting of stiffened chitin[15]. Their morphology is useful for taxonomic identifications, ranging from the order to the species level[16–23] (Fig. 2). Clarke[16], who first proposed the use of beaks in taxonomy, also pointed out the applicability of this approach in palaeontology. Since the pioneering work of Tanabe et al.[24], many fossil species of coleoids have been established based on beaks from Cretaceous sediments[23,25,26]. Further documentation of fossil beaks is, therefore, expected to reveal the evolutionary history of sepioids.

Here we report a new sepioid species based on fossil coleoid beaks from the Pierre Shale (~74 Ma) and Fox Hills Formation (~67 Ma), South

[1]Department of Earth and Planetary Sciences, Hokkaido University, Sapporo, Japan. [2]Spectroscopy and Imaging Division, Japan Synchrotron Radiation Research Institute, Sayo, Hyogo, Japan. [3]Department of Geosciences, Ruhr University Bochum, Bochum, Germany. [4]Morgenrot Inc., Tokyo, Japan. [5]Department of Geosciences, Osaka Metropolitan University, Osaka, Japan. [6]Department of Biological Sciences, Chuo University, Tokyo, Japan. [7]Brain Research Institute, Niigata University, Niigata, Japan. [8]American Museum of Natural History, New York, USA. [9]These authors contributed equally: Kanta Sugiura, Shin Ikegami. ✉ e-mail: iba@sci.hokudai.ac.jp

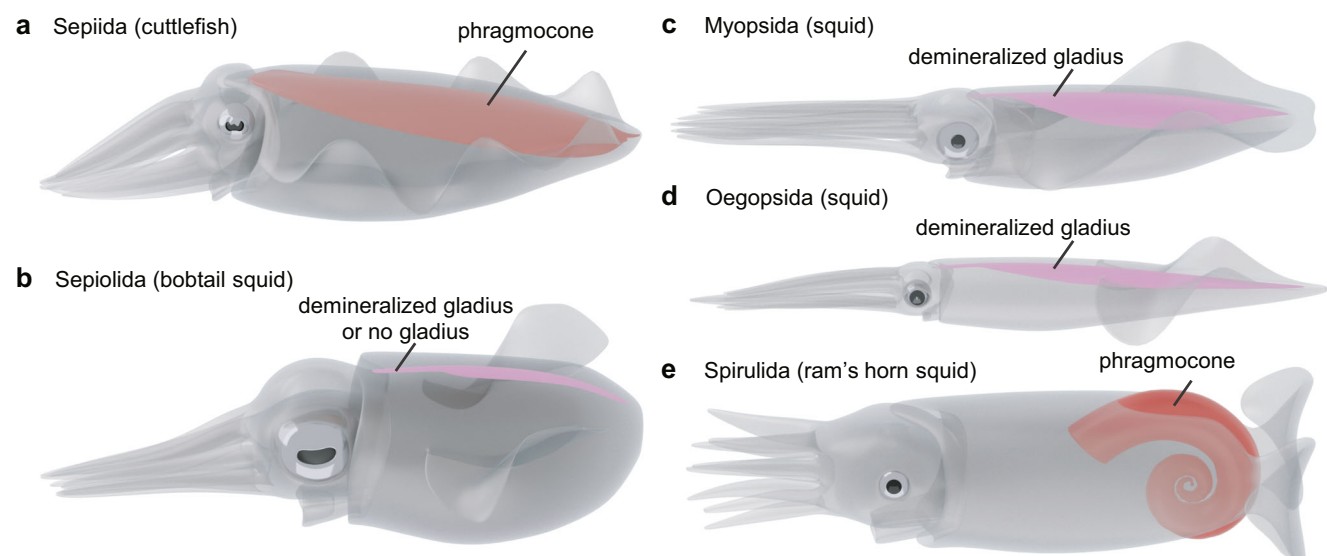

**Fig. 1 | Five orders of modern Decabrachia. a** Sepiida. **b** Sepiolida. **c** Myopsida. **d** Oegopsida. **e** Spirulida. Sepiida and Sepiolida are jointly combined as sepioids.

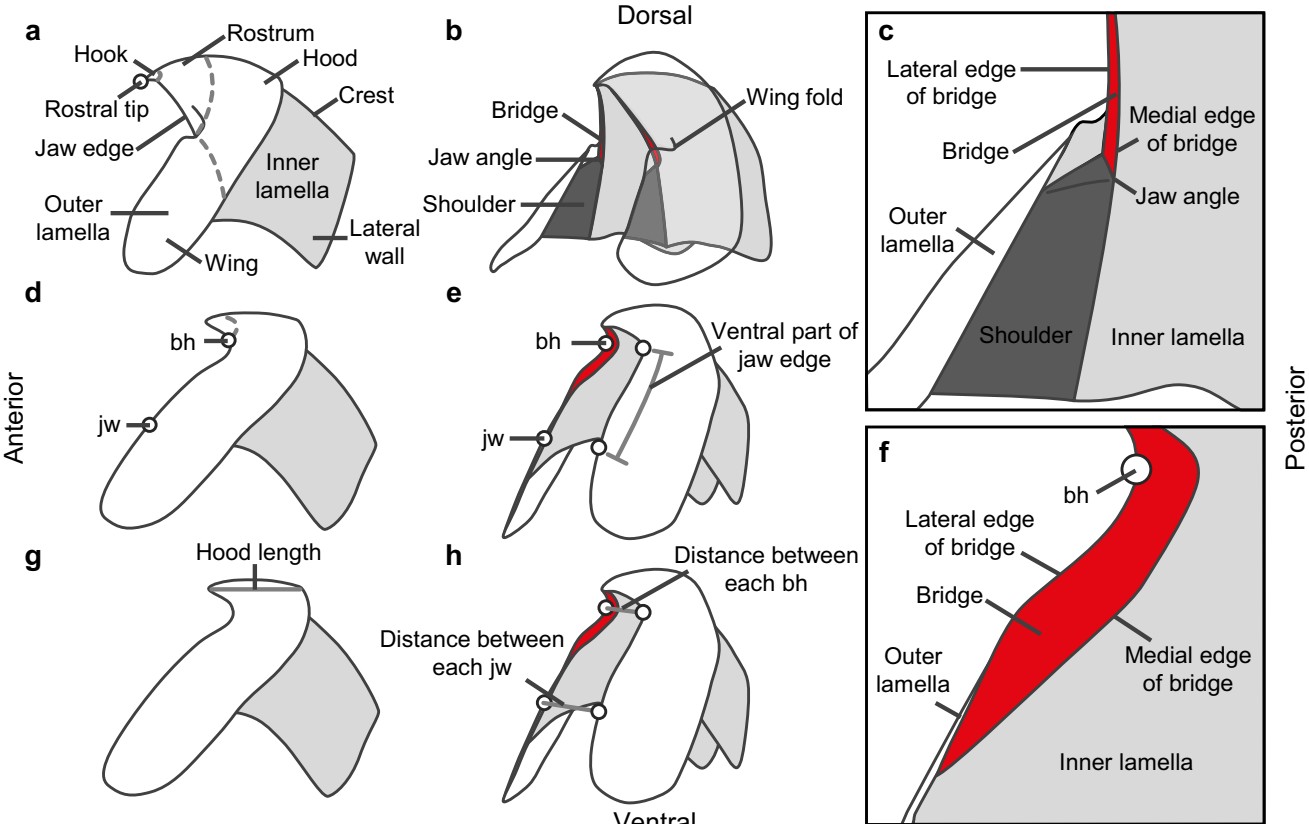

**Fig. 2 | Descriptive terms for the lower beak of coleoids. a–c** Terms for the morphology of the lower beak in coleoids. **d–f** Terms for the morphology specific to the sepioid lower beak. **g, h** Lengths measured in this study. **a, d, g** Lateral view. **b, e, h** Oblique view. **c, f** Magnified view of the bridge area. **a–c** are based on the lower beak of the Oegopsida, **d–h** are based on that of the Sepiida. Base of the hook (= bh), joint between jaw edge and anterior margin of wing (= jw), ventral part of jaw edge, and measurements in **h** are newly defined in this study. The other terms and measurements are based on published methods[17,23,34].

Dakota. These discoveries were facilitated by the Digital fossil-mining[23,27] incorporated with a zero-shot learning AI model (Fig. 3 and Supplementary Movie 1). The Digital fossil-mining method first converts whole rocks into high-resolution, full-coloured digital image datasets by using grinding tomography. Fossils in the datasets are then isolated from rocks through manual segmentation and are visualized as 3D models[23,27]. In this study, we replaced the human-based segmentation by a zero-shot learning AI model. Zero-shot learning AI enables the detection of any objects in images regardless of whether they are already known or unknown, without requiring an additional training process[28,29]. The Digital fossil-mining method, which incorporates zero-shot learning AI, is thus able to detect any fossils embedded in rocks even if they are taxonomically unknown (Fig. 3

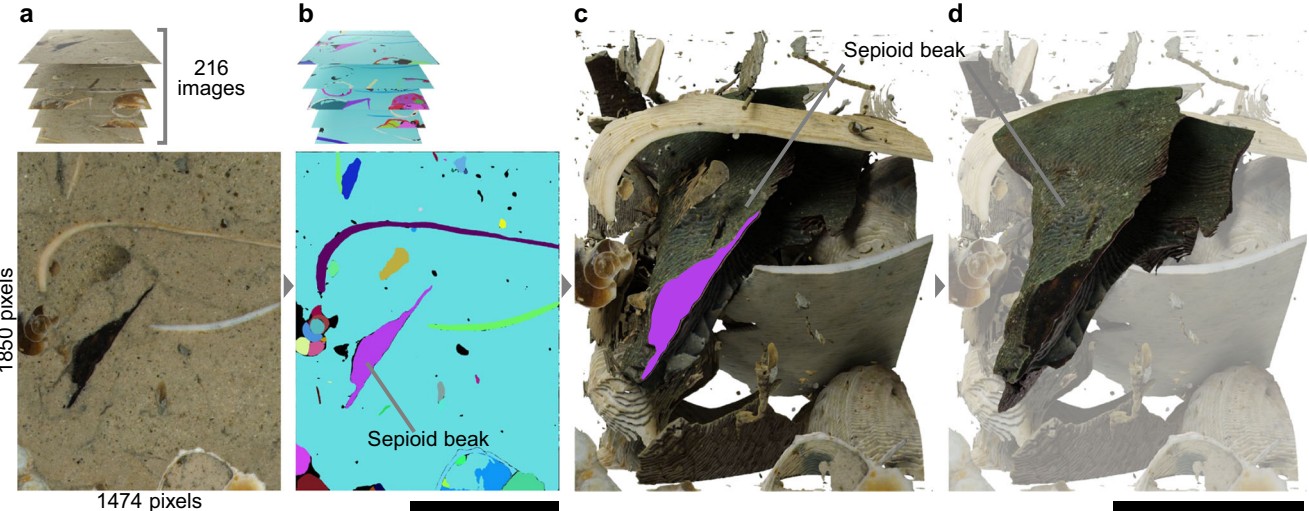

**Fig. 3 | The process of the Digital fossil-mining that incorporates a zero-shot learning AI. a** Original tomographic images of a Cretaceous carbonate concretion. **b** Segmentation by DEVA. **c** Visualization of all fossils as original-coloured 3D models. **d** The oldest sepioid lower beak (NMNS_DS00254_0k5cvkE.stl). Scale bars equal 5 mm. See also Supplementary Movie 1.

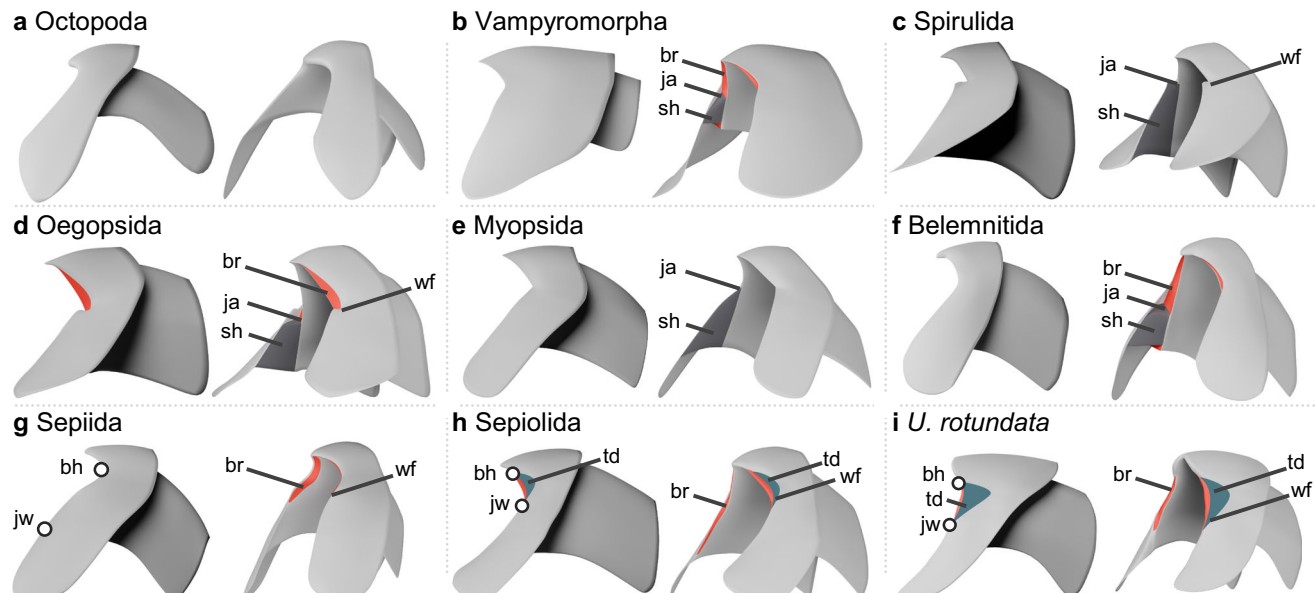

**Fig. 4 | Comparison of lower beak morphology of modern coleoid orders, Belemnitida, and *Uluciala rotundata* gen. et sp. nov. a** Octopoda. **b** Vampyromorpha. **c** Spirulida. **d** Oegopsida. **e** Myopsida. **f** Belemnitida. **g** Sepiida. **h** Sepiolida. **i** *U. rotundata* gen. et sp. nov. bh base of the hook, br bridge (red), ja jaw angle, jw joint between jaw edge and anterior margin of wing, sh shoulder, td triangular depression (green), wf wing fold.

and Supplementary Movie 1). The AI model used here is the decoupled video segmentation approach (DEVA) and was originally developed for annotating movies[30]. This study applied the DEVA technique for the first time to tomographic datasets, comprising two-dimensional images. DEVA can annotate any object without training because it is based on the Segment Anything Model, which has been pre-trained by over one billion generalized mask data[31]. We applied these methods to two carbonate concretions from sediments of the Western Interior Seaway, which formed a north–south aligned inland sea in North America during the Late Cretaceous. The sediments co-occurring with the examined concretions yield abundant well-preserved fossils, which provide a reliable, precise biostratigraphic scheme[32,33]. The carbonate concretions preserve the three-dimensional morphology of fossils, which is essential for precise beak identification[26]. Our digital methods allowed accurate visualization of the beak fossils in the concretions. This study shows that these beaks include the oldest sepioid ever discovered. It shows morphological features intermediate between the Sepiida and Sepiolida, providing a clue for the deep-time evolution of both groups.

## Results
### Systematic palaeontology
Subclass Coleoidea Bather, 1888
 Superorder Decabrachia Haeckel, 1866
 Order and Family uncertain

**Remarks**. The specimens discovered here are characterized by wide bridges. The inner edge of either bridge is directly connected to the anterior margin of the adjacent wing without forming jaw angles and shoulders. These specimens differ from taxonomically related groups as follows: Octopoda have no bridges[17] (Fig. 4a), Vampyromorpha show

distinct shoulders[17,34] (Fig. 4b), Spirulida, Oegopsida, and Myopsida possess clear jaw angles[17] (Fig. 4c–e). The lower beak morphology of the Belemnitida has been documented from two Early–Late Jurassic species *Hibolithes semisulcatus* and *Acrocoelites conoideus* (?)[35,36]. These fossils indicate the presence of a small rostral hook, a short hood, nearly straight jaw edges, distinct bridges that converge to the rostral tip, wings separated from the jaw edges, and the absence of pigmented shoulder parts[35,36] (Fig. 4f). The newly discovered specimens are distinguished from the belemnitid lower beak by their anterior margin of the wings, which is directly connected to the adjacent jaw edge (Fig. 4f). These distinctions are in common with those of the Sepiida and Sepiolida (Fig. 4g, h). The specimens described here have a large hook, and the ventral part of the jaw edge extends anteroventrally (Fig. 4i), features that are present in the Sepiida but not in the Sepiolida (Fig. 4g, h). By contrast, these specimens show a straight ventral part of the jaw edge and well-developed triangular depressions on both sides of the rostrum (Fig. 4i). Both features are seen in the Sepiolida but not in the Sepiida (Fig. 4g, h). We therefore interpret these lower beaks as morphologically intermediate between those of the Sepiida and Sepiolida.

*Uluciala* gen. nov.

**Etymology**. After the Latin *ulucus* (owl) and *ala* (wing), referring to the shape of the beak.

**Type species**. *Uluciala rotundata* sp. nov.

**Diagnosis**. As for the type species by monotypy.

*Uluciala rotundata* sp. nov.

**Etymology**. After the Latin *rotundata* (rounded), referring to the rounded rostrum.

**Material**. NMNS_DS00254_0k5cvkE.stl and NMNS_DS00285_ma2 jp6i.stl[37], lower beaks deposited in the National Museum of Nature and Science (NMNS), Tokyo, and the American Museum of Natural History (AMNH). These 3D models are also openly accessible from Figshare (https://doi.org/10.6084/m9.figshare.28119998)[37] and can be freely viewed through various types of software or 3D printing.

**Locality and horizon**. NMNS_DS00254_0k5cvkE.stl is from AMNH loc. 3274, Meade County, South Dakota, USA; *Baculites compressus/cuneatus* ammonite Zone (~74 Ma, middle upper Campanian), Pierre Shale[32]. NMNS_DS00285_ma2jp6i.stl is from AMNH loc. 3272, Dewey County, South Dakota, USA; *Hoploscaphites nebrascensis* ammonite Zone (~67 Ma, upper Maastrichtian), Timber Lake Member, Fox Hills Formation[38].

**Diagnosis**. The rostral tip forms a rounded, large hook. A deep groove extends from the base of the hook on both sides to halfway up the hood. This groove runs parallel to the crest in lateral view. Bridges gradually become indistinct towards the rostral tip. They are smoothly connected to the anterior margin of the adjacent wing without a shoulder in between. The medial edge of the bridge is anterior to the lateral edge. The ventral part of the jaw edges is straight in lateral view and extends anteroventrally. Wings extend anteroventrally and form a low wing fold. The rostrum has triangular depressions surrounded by the ventral part of the jaw edge, a wing fold, and a groove extending from the base of the hook. No folds or ridges exist on the lateral walls.

**Description**. NMNS_DS00254_0k5cvkE.stl is a small lower beak, 5.23 mm in hood length (Fig. 5a–g). It is well-preserved except for its left wing and the posterior end of the lateral walls (Fig. 5a–g). The rostrum reaches approximately half of the lateral walls in height (Fig. 5a, c, d). It is

as high as broad (Fig. 5b), with a distance of 5.66 mm between the point where the jaw edge and the anterior margin of the wing merge on both sides, and 6.32 mm between the rostral tip and the point where the jaw edge and the anterior margin of the wing merge. The rostral tip is rounded and hooked, forming a concave outline in dorsal view (Fig. 5a, c–e). The hook is large and broad (Fig. 5a–e), in which the distance between the base of the hook on both sides is ~0.46x the distance between the ventral ends of the lateral wall on both sides. The distance between the base of the hook and the rostral tip is 2.10 mm, and that between the base of the hook on both sides is 2.65 mm. The rostrum has two grooves on each side. The deeper groove extends from the base of the hook to halfway up the hood (Fig. 5a, c–e). It runs parallel to the crest in lateral view and is 2.29 mm in length (Fig. 5a, c, d). The shallower groove is located in the middle of the deeper groove and the crest, extending from the posterior end of the hood toward the rostral tip (Fig. 5a, c–e). It becomes indistinct anteriorly and disappears around the hook. It runs parallel to the deeper groove and is 4.59 mm in length (Fig. 5a, c–e). Bridges are broad and become indistinct toward the rostral tip (Fig. 5b, f, g). The medial edge of the bridge is anterior to the lateral edge. Jaw edges are smoothly connected to the anterior margin of the adjacent wing without shoulders in between (Fig. 5f, g). Jaw angles are absent. The ventral part of the jaw edges is straight in lateral view and extends anteroventrally (Fig. 5a, c, d). In lateral view, the ventral part of the jaw edges forms an angle of 74.7° with the line where the hood length is measured. Wings extend anteroventrally and form a low wing fold (Fig. 5a–g). The rostrum has triangular depressions on both sides (Fig. 5a, c, d). These depressions are surrounded by the ventral part of a jaw edge, a wing fold, and a deeper groove. No folds or ridges exist on the lateral wall (Fig. 5a, c, d). The pigmented part of the wings extends more ventrally than the lateral walls (Fig. 5a, c, d). In the pigmented part, the lateral walls are twice higher in their anterior part than in their posterior end (Fig. 5a, c, d).

NMNS_DS00285_ma2jp6i.stl is a small lower beak, well-preserved except for the wings and the posterior part of the lateral walls (Fig. 5h–l). It shows the same diagnostic characters as NMNS_DS00254_0k5cvkE.stl (Fig. 5h–l). Shallower grooves are not observed. This character has probably been lost in this specimen since its hood is preserved as a mould (Fig. 5h–l). Measurements of length for this specimen are as follows; hood length, 6.77 mm; distance between the point where the jaw edge and the anterior margin of the wing merge on both sides, 8.21 mm; distance between the rostral tip and the point where the jaw edge and the anterior margin of the wing merge, 7.88 mm; distance between the base of the hook and the rostral tip, 2.51 mm; distance between the base of the hook on both sides, 3.46 mm; groove length; 2.54 mm. The distance between the base of the hook on both sides is ~0.41x the distance between the ventral end of the lateral wall on both sides. The ventral part of the jaw edges form an angle of 71.8° with the line where the hood length is measured.

## Discussion
### Discovery of the oldest sepioid
The morphology of chitinous cephalopod beaks strongly correlates with their phylogenetic relationships rather than with feeding habits[39,40]. The beaks are used for taxonomic identifications from the order to the species level[16–23]. Only the lower beak was used here for the taxonomic identification since phylogenetic differences are much less obvious in the upper beak[17,34]. The fossil record of coleoids without or only weakly sclerotized internal skeleton is, apart from several reports of their beaks and statoliths, mainly restricted to imprints of the soft body or gladius preserved in Fossil Lagerstätten[13]. This limitation of the record causes a serious bias in the chronological and geographical distribution of fossil Decabrachia. Fossil lower beaks from non-Lagerstätten sites provide an important source of data contributing to the understanding of their deep-time evolution, which has not been solved by molecular data, phragmocones, or gladii.

The Western Interior Seaway was a huge inland sea, the region showing the highest cephalopod diversity (e.g., ammonites) in the Late Cretaceous[41].

Since 1856, abundant phragmocones, rostra, and gladii of fossil coleoids have been reported from this region[42,43]. These fossils have been assigned to two taxonomic orders, the Belemnitida and the Octopoda. The Belemnitida is an extinct internally shelled group of Decabrachia that flourished from the Late Triassic to Cretaceous[44]. The belemnitids that have been discovered in sediments of early Cenomanian to late Maastrichtian age (100–66 Ma) in the Western Interior Seaway consist of 7 species attributed to 3 genera (*Bellemnitella*, *Neohiblites*, *Praeactinocamax*)[42,43,45–48]. The Octopoda,

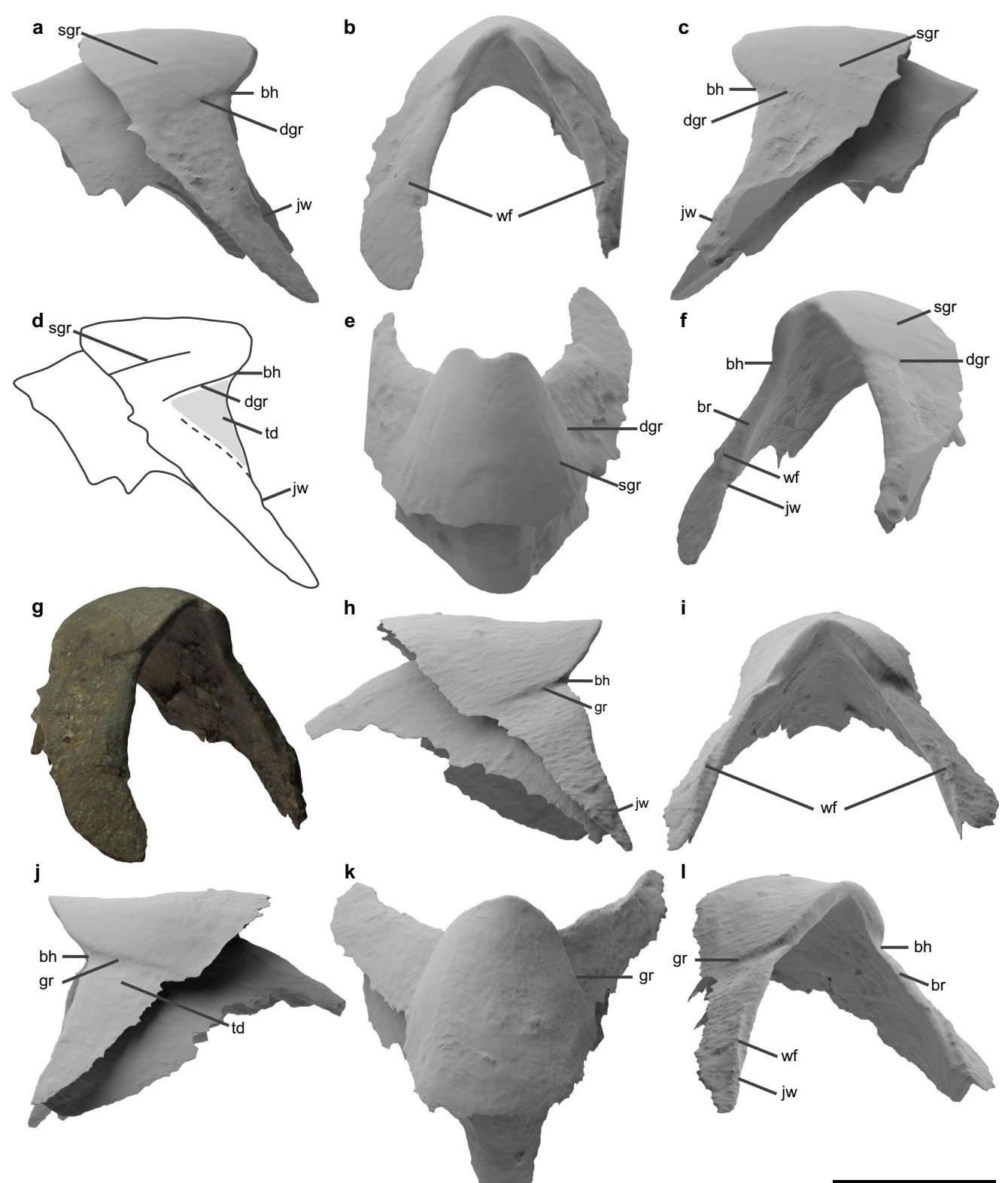

**Fig. 5 | Lower beaks of *Uluciala rotundata* gen. et sp. nov.**
**a**–**g** NMNS_DS00254_0k5cvkE.stl, **h**–**l** NMNS_DS00285_ma2jp6i.stl. **a, h** Left lateral view. **b, i** Anterior view. **c, j** Right lateral view. **d** Schematic drawing of (**a**), showing a triangular depression in grey. **e, k** Dorsal view. **f** Right oblique view. **g** Left oblique view in the original colours. **l** Left oblique view. bh base of the hook, br bridge, dgr deeper groove, gr groove, jw joint between the jaw edge and the anterior margin of the wing, sgr shallower groove. td triangular depression, wf wing fold. Scale bar equals 5 mm.

represented by extant octopuses, is a gladius-bearing or soft-bodied group included in the superorder Octobrachia[49]. The fossils of the Octopoda from the Western Interior Seaway, all large gladii, reach lengths of more than 1 m and range stratigraphically from the middle Turonian to the late Maastrichtian (93–66 Ma)[48]. A total of 8 species assigned to 5 genera (*Actinosepia, Enchoteuthis, Muensterella, Niobrarateuthis, Tusoteuthis*), which all belong to stem groups of the octopuses (suborder Teudopseina), have been documented from the fossil record[48–50]. *Uluciala rotundata* gen. et sp. nov., which is based on fossil beaks differs from both the Belemnitida and the Octopoda on the order level. Its lower beak morphology is clearly distinguished from that of the Octopoda by the presence of distinct bridges and from that of the Belemnitida by the anterior edge of the wings directly connected to a bridge. In the past two fossil coleoid beaks have been described from this region[32,51], but both are upper beaks, not suited for a taxonomic classification[17,34]. Our findings thus indicate the presence of a coleoid group hitherto unknown from the late Campanian to the late Maastrichtian of the Western Interior Seaway. This is therefore the first record of non-belemnitid decabrachians from the Western Interior Seaway.

The lower beak morphology, such as the broad hook and depressions on the rostrum, indicates that *Uluciala rotundata* gen. et sp. nov. had an intermediate morphology between the Sepiida and Sepiolida (Figs. 4 and 5). This result is well-supported by morphological disparity analyses of *U. rotundata* and modern cephalopod species (Supplementary Discussion and Supplementary Figs. 1 and 2). Our findings suggest that *U. rotundata* is most likely a taxon showing the process of differentiation between the two orders. The Sepiida have a large internal mineralized phragmocone (cuttlebone: Fig. 1a)[3]. Modern Sepiida thrive globally in the shallow areas of oceans except along the coasts of North and South America, providing a large biomass[7]. Currently, the oldest fossil of the Sepiida is a cuttlebone from the upper Maastrichtian (*Ceratisepia vanknippenbergi*; Maastricht Formation, ~70 Ma) of the Netherlands[9]. So far it is the only known Mesozoic specimen of sepioids. In the Cenozoic, the Sepiida have a chronologically continuous fossil record from the Palaeocene to Holocene[52]. The Sepiolida have a significantly reduced gladius or are even completely soft-bodied (Fig. 1b)[5]. Representatives of this group are typically smaller than 100 mm, distributed worldwide, and serve as model animals in genomics and neurobiology[5,53]. The Sepiolida have no reliable fossil record so far[11,12], making their deep-time evolution completely unknown. Here, we describe a specimen of *U. rotundata* from the late Campanian as the stratigraphically oldest representative of the sepioids.

The new finding described here was facilitated by the first application of a zero-shot learning AI model both to palaeontological studies and to full-coloured tomographic datasets (Fig. 3). Recently, AI models have been adopted for the segmentation of fossils in tomographic datasets[54–57]. These models required considerable effort to generate training data and conduct deep learning. Despite such efforts, these models cannot be used for detecting fossils of unknown taxa because they were developed for specific fossils. Our methods are, in contrast, able to excavate any fossils from tomographic datasets without requiring training data (Fig. 3b, c). These methods facilitate discoveries of unexpected fossils, including unknown taxa, and consequently accelerate our understanding of palaeobiodiversity (Fig. 3).

## Implications for sepioid evolution

Our findings provide significant insights into the evolution of the Sepiida and Sepiolida. Previous studies have proposed different phylogenetic relations of the Decabrachia[58–61]. Modern Decabrachia show numerous possibly convergent characteristics at the order level, induced by their rapid radiation[14]. It is, therefore, difficult to reconstruct their phylogenetic relationships only based on extant species, even with the extensive morphological and molecular data available. Tanner et al.[59] analyzed mitochondrial protein-coding genes of 19 species, and positioned the two orders as adjacent branches. Contrary to that, Anderson and Lindgren[61] used transcriptome assemblies of 31 species, placing the Sepiida as the most derived clade and the Sepiolida as a distant branch. Morphological data of beaks are

also insufficient to reconstruct the decabrachian phylogenetic tree in a resilient way, if based on extant species only[39]. Constraints based on chronologically successive fossil records are required to solve this problem, for both sepioids and non-sepioids[62]. Nevertheless, the disparity analyses conducted here show morphological similarity among taxa[63], which in beaks can be regarded as phylogenetic distance[26]. The discovery of *Uluciala rotundata* gen. et sp. nov. provides evidence that the Sepiida and Sepiolida are closely related to each other. The finding indicates that the sepioids were at an early diverging stage in the later part of the Late Cretaceous, and the Sepiida and Sepiolida evolved their modern beak forms thereafter.

## Methods
### Geological settings

Two cephalopod beaks (NMNS_DS00254_0k5cvkE.stl and NMNS_DS00285_ma2jp6i.stl) were retrieved from the Western Interior seaway deposits. NMNS_DS00254_0k5cvkE.stl was from the upper part of the Pierre Shale in Meade County, South Dakota, AMNH loc. 3274[32]. It was embedded in a carbonate concretion from the *Baculites compressus/cuneatus* ammonite Zone, indicating a late Campanian age (~74 Ma)[33]. NMNS_DS00285_ma2jp6i.stl was from Timber Lake Member, Fox Hills Formation in Dewey County, South Dakota, AMNH loc. 3272[38]. It was embedded in a carbonate concretion from the *Hoploscaphites nebrascensis* ammonite Zone, indicating a late Maastrichtian age (~67 Ma)[38].

### The Digital fossil-mining method combined with a zero-shot learning AI

In a first step, carbonate concretions were cut into blocks. These blocks were then converted into 2970 and 2638 cross-sectional images with high-resolution and original colours by using grinding tomography[64–66]. These images have a dimension of 19,008 × 12,672 pixels, and the interval between each image is either 50 μm or 25 μm (Supplementary Data 1). Details of the grinding tomography system (Palaeobiology Lab, Hokkaido University) used in this study are described in previous studies[23,67–69].

We applied DEVA (decoupled video segmentation approach)[30] to a cropped image dataset of the concretion (Fig. 3a, b). DEVA was run on a Supermicro AS -4125GS-TNRT server. This unit is equipped with two CPUs (AMD EPYC 9374F, 32 cores, base clock 3.85 GHz) and a GPU (NVIDIA RTX 6000 Ada) mounted on a motherboard (Supermicro Super H13DSG-O-CPU), 1.5 TB RAM configured with 24 modules of 64GB DDR5 (Micron MTC40F2046S1RC48BA1), and Ubuntu 22.04 through WSL2 on Microsoft Windows 11 Pro. DEVA was run with the following script: python3 demo/demo_automatic.py --chunk_size 4 --img_path ##*path_for_input_files*## --temporal_setting semionline --size -1 --output ##*path_for_output_files*## --SAM_NUM_POINTS_PER_SIDE 64 --max_num_objects -1. All the data from DEVA were visually checked to ensure that the fossil beak area is properly masked. We then imported the mask data from DEVA to Amira 3D v2023.2. (Thermo Fisher Scientific), unified masks of the lower beak area into a single label using the Convert Image Type function and Magic wand tool. The beaks were subsequently visualized as 3D models (Fig. 3c, d). The 3D models underwent minimal smoothing consistent with imported masks to avoid modifying the diagnostic characters. Measurements were performed using Amira and ImageJ v1.54g[70]. Visualizations for the figures were created using Blender v4.3.2 (Blender Foundation). In Supplementary Data 1, we documented the details of the original tomographic datasets of carbonate concretions, the reconstructed 3D model of the fossil sepioid beaks, the label data corresponding to each model, and the cross-sectional images cropped for generating the label data.

### Taxonomy

In this study, the new taxon was identified based on the systematic classification. The terms and measurements used here to describe the morphology of the beaks follow published terminology[17,23,34] (Fig. 2a–c, g). Additionally, we define the following beak characters (Fig. 2d–f, h): the joint between the jaw edge and the anterior margin of the wing, base of the hook,

and the ventral part of the jaw edge. The base of the hook is the point on the jaw edges showing the strongest curvature. The ventral part of the jaw edge is the extension from the base of the hook to the joint between the jaw edge and the anterior margin of the wing. These new terms are helpful for describing beaks of the Sepiida and Sepiolida, in which rounded jaw angles do not form a distinct point (Fig. 4). We measured the distance between the base of the hook on both sides, and that between the joint between the jaw edge and the anterior margin of the wing on both sides (Fig. 2h).

### Nomenclatural acts

This published work and the nomenclatural acts it contains have been registered in ZooBank, the proposed online registration system for the International Code of Zoological Nomenclature. The ZooBank LSIDs (Life Science Identifiers) can be resolved and the associated information viewed through any standard web browser by appending the LSID to the prefix "https://zoobank.org/". The LSIDs for this publication are: E6725C0A-4BFD-44F9-B53F-C4D3679C8CFC for the new genus *Uluciala*, and A5A7AE5C-503A-4EB8-BFCC-804E76B0A7A2 for the new species *U. rotundata*.

The fossils described here are thin, small, and were embedded in hard rocks. It was, therefore, impossible to document and reliably diagnose them without using destructive methods. We therefore follow Declaration 45 and Recommendation 73G–J of the International Code of Zoological Nomenclature[71], which allows the establishment of new species without designating a type specimen[72]. To satisfy Recommendation 73 J of providing "*extensive documentation of potentially diagnostic characters as completely as possible*" (International Commission on Zoological Nomenclature 2017, p. 96)[71], we archived all digital data of fossil beaks in the National Museum of Nature and Science (NMNS), Tokyo. These data include the 3D models, mask data, cropped cross-sectional images, and original cross-sectional images with their raw data. Accession numbers of these data are provided in Supplementary Data 1. A copy of these data is deposited in the Division of Paleontology, the American Museum of Natural History (AMNH), New York. The 3D models, mask data, and cropped cross-sectional images are also available in Figshare (https://doi.org/10.6084/m9.figshare.28119998)[37]. As these data are given in universal formats, they can be freely viewed through various types of software or 3D printing.

### Morphological disparity analyses of the cephalopod lower beaks

To validate our systematic classification, we evaluated the morphological disparity of the lower beak morphology of modern cephalopod species and the new species described here. This analysis is based on Gower's similarity coefficient and a principal coordinates analysis (Supplementary Discussion and Supplementary Figs. 1 and 2). The data of the modern beaks were collected from 157 species in the most comprehensive database[73] and 8 species stored in NMNS. These taxa cover all extant orders and >90% of the extant cephalopod families[7]. The morphological data consist of 46 characters, which are tabulated in a character matrix. Gower's similarity coefficient was calculated based on this matrix, and the generated distance matrix was then used in the principal coordinate analysis. We used R 4.4.2 (the R Foundation) for these calculations. The data and R code for this analysis are published in Figshare (https://doi.org/10.6084/m9.figshare.29665496)[74].

### Reporting summary

Further information on research design is available in the Nature Portfolio Reporting Summary linked to this article.

### Data availability

The 3D models, the label data, and the cropped cross-sectional images of the discovered beaks are available in Figshare (https://doi.org/10.6084/m9.figshare.28119998)[37]. All tomographic images of the carbonate concretions (~5 TB) are archived in the National Museum of Nature and Science, Tokyo, and the American Museum of Natural History, New York. All data related

to this paper are available from the corresponding author upon reasonable request.

### Code availability

The data and R code used for morphological disparity analyses are available in Figshare (https://doi.org/10.6084/m9.figshare.29665496)[74].

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

## Acknowledgements

We thank Makoto Manabe, Chisako Sakata, Aya Togashi, Emi Kosugi, Hyu Ito, Ami Koshimizu, and Reiko Nozaki for archiving the data; Hidehiko Nomura and Kosuke Nakamura for preparation of specimens; Tsuyoshi Ishii, Masashi Takei, and Shintaro Sasaki for assistance with the operation of the grinding tomography; Bushra Hussaini and Anastasia Rashkova for giving access to the studied concretion in museum collections; Kazunori Hasegawa, Keisuke Nakamoto, and Takeya Moritaki for access to modern cephalopod beaks in museum collections; and Yuzuru Ikeda for the critical discussion. This work was supported by the Japan Society for the Promotion of Science (grant nos. 22J13936 for S.I., 23K17274 for Y.T., 19H02010 for Y.I., 22H02937 for K.T., 23H02544 for H.N. and A.K., 25K22459 for Y.I.), Japan Aerospace Exploration Agency (grant no. JX-PSPC-540452 for Y.I., M.O.D., and T.H.), Chuo University Grant for Special Research 2022–2023 (H.N.), Chuo University Joint Research Grant (H.N., A.K., and Y.I.), and Ami Koshimizu Research Grant 2025 (Y.I.).

## Author contributions

K.S., S.I., Y.T., and Y.I. conceived this project. Y.I. supervised this project. S.I., N.H.L., and Y.I. collected samples. K.S., S.I., Y.T., M.O.D., and Y.I. developed the methods with contributions from A.K., H.N., K.T., and T.H. K.S. and S.I. curated the datasets. K.S. and S.I. performed taxonomic analysis. K.S. and S.I. conducted a 3D visualization of fossil beaks. S.I., Y.T., J.M., and Y.I. verified the reproducibility of the results. S.I., Y.T., M.O.D., A.K., H.N., K.T., T.H., and Y.I. acquired funding. K.S., S.I., J.M., and Y.I. wrote the manuscript. All authors edited and approved the manuscript.

## Competing interests

The authors declare no competing interests.
