## [Transparent Peer Review file · Communications Biology]

The oldest sepioid cephalopod from the Cretaceous discovered by Digital fossil-mining with zero-shot learning AI

Corresponding Author: Dr Yasuhiro Iba

Version 0:

Reviewer comments:

Reviewer #1

(Remarks to the Author)

Dear authors,

In general, I find your manuscript well organized, your methods the core of this contribution, your descriptions concise (though in places difficult to comprehend), and your conclusions reasonable. Your figure may be larger.

I tentatively agree with your systematic assignment. However, the morphologic differentiation of *Uliciala rotundata* n. sp. is - in my opinion - developable; you focus on comparisons with modern sepiolids and sepiids; comparisons with co-existing coleoids are missing (Are there more records of beaks in Western Interior sediments?). Especially belemnitids, which are commonly seen as stem decabrachians and thus as precursors of sepiids and sepiolids, should be considered. Admittedly, belemnite beak morphology is poorly known, but a brief review that exposes differences is recommended here.

The ms includes a phylogenetic implications section, which is likewise developable. In order to reach a wider audience, perspective questions should be addressed: How does the discovery of *Uliciala rotundata* n. sp. impacts previous ideas on the origin and especially the ambiguous monophyly of the Sepioidea? Does your discovery influences the systematic position of contemporary *Ceratisepia*? Are there any implications addressing the evolution/origin of the sepioid cuttlebone/*gladius*?

All in all, I agree with your careful formulation in the end (l. 261) "...*Uliciala rotundata* n. sp. provides evidence that Sepiida and Sepiolida are closely related....".

After minor revisions (see my detailed annotations in the enclosed pdf) the ms should be considered for publication.

Dirk Fuchs

Reviewer #2

(Remarks to the Author)

Overview.

This paper describes a new genus and species of sepioid cephalopod from the Cretaceous. This effort is a novel use of zero-shot learning (ZSL) on fossil data. This new genus and species sheds light upon the deep time history of this group and aids in understanding the timing of divergence between the Sepiida and Sepiolida, especially for groups with limited fossil records.

Major Comments.

The paper provides minimal background on ZSL and provides minimal context for its use here. To engage the paleo community, it would help to discuss the advantages and drawbacks of this method, especially compared to other machine learning approaches. Was the approach attempted on a CT scan of the block before taking the serial sectioning approach?

Further, the discussion passage L241-249 might be misleading. As far as I can tell, the deep learning approach in this paper is for segmentation, not taxonomic identification. The latter would involve incorporating beaks for other taxa to evaluate proximity of the unknown object in the known, taxonomic embedding space. Thus, the usage appears to be more about

object detection, not classification per se. I would hesitate to describe zero-shot as capable of discovering unknown fossils in a taxonomic sense without this added context, but rather as useful for discovering objects that *may* be new taxonomic discoveries.

It's good to have the specs of the computing system used to fit the model (L285+), but it would be better to specify the arguments used for the run. Was the text-prompted mode for automatic mode used for DEVA? Were any non-standard arguments specified for the prompt? How was quality control and cross-checking of the segmentation output handled? Visual inspection? Were the DEVA segmentation masks for the beak manually adjusted in Amira?

More details on model results, any code, model tweaks, etc. would be critical for reproducing the results and allowing future applications of this method.

Fig. 5 shows many views of the beak for the new taxon, which is helpful for visualizing its morphology. Does the limited view of beak in Fig. 4 show the full differences in morphology among the taxa, or would additional views help here?

Line Comments.

L29: "in which an AI model has led to"

L57: switch "only" and "exceeded"

L83: delete "an"

L89: delete the commas around "thus." Also, it may to better connect the steps here as: "The system is thus able to isolate cohesive objects in matrix that may fossilized remains of unknown taxa."

L90 & 91: delete "old" after both instances of Ma

L91-92: what was the previous oldest known sepioid? Where was it from?

L107: should this be "the inner edge of either bridge is connected...?"

L109: delete comma after Octopoda

L144: delete "the" before "Timber Lake Member"

L201: change "on" to "of"

L229-230: suggest: "thrives globally in the shallow areas of all oceans except along the coasts of North and South America"

L236: add "serve as" between "and" and "model"

L256-257: I would reword as "even with the extensive morphological and molecular data available"

L263: Add an "and" between "Cretaceous" and "Sepiida"

L272 & 275: Does Zone need to be capitalized?

L282: Add an "and" between "pixels" and "the"

L322: Remove commas around "therefore"

Reviewer #3

(Remarks to the Author)

I co-reviewed this manuscript with one of the reviewers who provided the listed reports. This is part of the Communications Biology initiative to facilitate training in peer review and to provide appropriate recognition for Early Career Researchers who co-review manuscripts.

Reviewer #4

(Remarks to the Author)

Dear authors,

It is a great idea to apply grinding tomography to find fragile fossil structures such as non-mineralized cephalopod beaks. In that sense, I very much welcome your manuscript!

Additionally, the use of AI is novel.

Further, new occurrences of such cephalopod beaks are important because they are rare and fill big gaps in our knowledge. The problem with coleoid beaks is that the morphology differs rather subtly between taxa, making their identification difficult. I made several annotations in the pdf and highlight my main points of criticism here:

1. I am not convinced that this is a sepiid beak. For a sepiolid, it is on the giant side and to me, it does not look intermediate between the two sepiid clades but rather like lying phylogenetically outside. This shows a fundamental problem: How do you place the new species in systematics? You could use phylogenetic (Bayesian, max. likelihood) or morphometric methods but there is neither mentioned in the main text, but it should be. Could it be that it is a form in the stem of the two groups? This would make most sense to me, presuming your assignment to that clade holds true (I am not sure about this, though).
2. In places it sounds like you were the first using grinding tomography, which is not true. My work-group has published a series of papers employing this method to cephalopod conchs (Tajika et al. 2015, 2018, Naglik et al. 2015, 2016) and there are older papers on bivalves (Götz 2003) and even older ones by Stensiö or so, because before CT became standard, vertebrate, coral and brachiopod workers used this technology, although by hand. I think this should be properly credited.
3. The figure explaining the terminology is hard to understand.
4. Conclusions are provided before the discussion in lines 91-93. I would remove this and limit such interpretations to after the description and comparison.
5. The comparison is limited (at least visually in Fig. 4) to sepiids. I find that the beaks also resemble Ommastrephes, but maybe I have missed some detail. For the reader, it would be great if representative lower jaws of other decabrachian clades were included in Fig. 4 and not just their lateral aspects. Maybe you are right and I am wrong, but I am just not convinced about the systematic assignment.
6. For sepiolids, the beaks would be on the giant side, how do you explain this?

7. Are there any remains of upper beaks or gladius remains? In such early forms, I would expect that they still had some internal sclerotized support with fossilization potential comparable to beaks. This should be shortly discussed in either case.

The other comments are of secondary importance but should still be looked at.

Best regards,
Christian Klug

Version 1:

Reviewer comments:

Reviewer #1

(Remarks to the Author)

Dear authors, you adopted the main points the referees recommended in their first review. This improved the ms significantly.

Moreover, you supported your systematic interpretation by additional statistical analyses. Although I agree with your decision of an open nomenclature and although I still think your evolutionary/phylogenetic conclusions are immature, I recommend to accept the present ms as it is.

Dirk Fuchs

Reviewer #4

(Remarks to the Author)

Dear Yasu and co-authors,

I am happy with the ms now!

I look forward to see it published!

Best wishes,
Christian

Reviewer #5

(Remarks to the Author)

This article uses a zero-shot learning model to process and segment fossil images, separating the fossils from the surrounding rocks and constructing a three-dimensional model of the fossils. Eventually, an important new species was discovered. This article combines AI methods with paleontology, which is a typical interdisciplinary achievement and is worthy of publication.

Major issue:

An artificial intelligence model using zero-shot learning discovered a new type of fossil species. This statement is ambiguous, including the title of the article. In fact, the authors merely used the zero-shot learning method to process fossil images and separate them from the surrounding rocks. The zero-shot learning model in this article mainly played the roles of image processing and segmentation, rather than directly participating in the discovery and classification of new species. The discovery of the new species was not based on this model, but was rather determined by human analysis of the fossil's characteristics.

Minor issue

Lines 292-293 AI models have also been adopted in obtaining morphological characters from fossils, e.g. Liu et al 2024 Nature Ecology & Evolution

Line 330 "converted into ~3,000 cross-sectional images", here needs a precise number of images

Version 2:

Reviewer comments:

Reviewer #5

(Remarks to the Author)

I am happy with the updated MS. It is a nice work. I look forward to see it published.

Haijun Song

Point-by-Point Response to the Reviewers

COMMSBIO-25-4252

We sincerely appreciate the reviewers' valuable and insightful comments on our manuscript. We have accepted all the major suggestions and comments made by the reviewers, thereby improving the revised manuscript.

In the point-by-point response below (P. 2–22), the comments from the reviewers are underlined, our response is given in **blue color**. The final adjustments are all integrated in the Article File and are highlighted in **red color**.

The comments and responses are ordered as follows: Comments from Reviewer 1 (Remarks to the Author, P. 2–3; Attached PDF, P. 4–7), Comments from Reviewer 2 (Remarks to the Author, P. 8–11), Comments from Reviewer 3 (Remarks to the Author, P. 12), Comments from Reviewer 4 (Remarks to the Author, P. 13–15; Attached PDF, P. 16–22).

Comments from Reviewer 1 (Remarks to the Author)

Dear authors,

In general, I find your manuscript well organized, your methods the core of this contribution, your descriptions concise (though in places difficult to comprehend), and your conclusions reasonable.

We thank Reviewer 1 for all the positive comments and for carefully checking our manuscript.

Your figure may be larger.

Following the suggestion, we enlarged Figs. 1, 2, 3, and 4 (P. 3, 4, 6, and 8 in the Article File).

I tentatively agree with your systematic assignment. However, the morphologic differentiation of *Uluciala rotundata* n. sp. is - in my opinion - developable; you focus on comparisons with modern sepiolids and sepiids; comparisons with co-existing coleoids are missing

Following the suggestion, we added a detailed description of the co-existing coleoid orders, Belemnitida and Octopoda (P. 14, L. 263–269 in the Article File). We also included images of the lower beak morphology of non-sepioid coleoid groups in the revised Fig. 4 for comparison (P.8 in the Article File). We further conducted the morphological disparity analysis of the lower beak for *Uluciala* and 165 modern cephalopod species, which cover all extant orders and >90% of all modern cephalopod families. The details of the analysis and the results are shown in the Supplementary Discussion and Supplementary Figs. 1 and 2 (P. 2–4 in the Supplementary Information File). In short, *Uluciala* is more closely related to the Sepiida and Sepiolida than to any other coleoid group and is therefore placed at an intermediate position between the two orders. These results support our systematic assignment.

(Are there more records of beaks in Western Interior sediments?).

Two fossil records of coleoid beaks are known from the Western Interior Seaway (Landman and Klofak 2012, Fig. 6E; Landman et al. 2015, Fig. 6b–d). These are upper beaks, which cannot be used for taxonomic identification. We added an explanation about these fossil beaks in the revised manuscript as follows: “In the past two coleoid beak fossils have been described from this region^{33,51}, but both are upper beaks, not suited for a taxonomic classification^{16,35}.” (P. 14, L. 267–269 in the Article File).

References;

Landman, N. H. & Klofak, S. M. Anatomy of a concretion: life, death, and burial in the Western Interior Seaway. *Palaios* **27**, 671–692 (2012).

Landman, N. H., Grier, J. C., Grier, J. W., Cochran, J. K. & Klofak, S. M. 3-D orientation and distribution of ammonites in a concretion from the Upper Cretaceous Pierre Shale of Montana. *Swiss J. Palaeontol.* **134**, 257–279 (2015).

Especially belemnitids, which are commonly seen as stem decabrachians and thus as precursors of sepiids and sepiolids, should be considered. Admittedly, belemnite beak morphology is poorly known, but a brief review that exposes differences is recommended here.

Following this suggestion, we added a brief review of the lower beak morphology of the order Belemnitida in the revised manuscript (P. 6–7, L. 123–130 in the Article File). This change clarifies the differences between Belemnitida and *Uluciala*.

The ms includes a phylogenetic implications section, which is likewise developable. In order to reach a wider audience, perspective questions should be addressed: How does the discovery of *Uliciala rotundata* n. sp. impacts previous ideas on the origin and especially the ambiguous monophyly of the Sepioidea?

Our findings are crucial in providing effective tools, namely fossil beaks, for understanding the so far unsolved origin and phylogeny of the sepioids. *Uliciala rotundata* alone does not constrain the monophyly of sepioids. Further documentation of fossil sepioid beaks is, however, expected to reveal their ambiguous evolutionary history. We changed the sentence about the significance of the fossil beaks in the revised manuscript as follows: “Fossil lower beaks from non-Lagerstätten sites provide an important source of data contributing to understand their deep-time evolution, which has not been solved by molecular data, phragmocones, or gladii.” (P. 13, L. 244–246 in the Article File).

Does your discovery influences the systematic position of contemporary *Ceratisepia*?

The characteristics of *Ceratisepia* clearly indicate that it belongs to the Sepiida. Our discovery thus does not change the systematic position of *Ceratisepia*.

Are there any implications addressing the evolution/origin of the sepioid cuttlebone/gladius?

Uliciala rotundata gen. et sp. nov. might have already demineralized its shell since no sepioid phragmocone fossils are known from the Western Interior Seaway. However, we avoided adding this speculative discussion since the evolution of beaks does not directly correlate with that of the cuttlebone/gladius.

All in all, I agree with your careful formulation in the end (l. 261) “...*Uliciala rotundata* n. sp. provides evidence that Sepiida and Sepiolida are closely related....”.

After minor revisions (see my detailed annotations in the enclosed pdf) the ms should be considered for publication.

Dirk Fuchs

We thank Reviewer 1 for all the positive comments and insightful feedback on this work. We added point-by-point responses for the comments in the Attached PDF on the following pages.

Yasu Iba

Comments from Reviewer 1 (Attached PDF)

P. 3, Fig. 1: "shell" is a very general term (compared to "gladius"); I recommend to use "phragmocone"

Following the suggestion, we changed “shell” to “phragmocone” in the revised Fig. 1 (P. 3 in the Article File). We also modified the explanation of the sepiid phragmocone and sepiolid demineralized gladius to be more precise in the revised manuscript (P. 2, L. 41–43 in the Article File).

P. 3, L. 53: Thousands of fossil cuttlebones do exist from various stratigraphic levels all over the world; hence, the evolution of the cuttlebone is comparatively well understood (as can be collected from a whole bunch of literature)!

As Reviewer 1 pointed out, fossil cuttlebones have abundant records from Cenozoic sediments. In contrast, so far only one specimen has been described from Mesozoic sediments (Hewitt and Jagt, 1999). Thus, the early evolution of the Sepiida, especially their phylogenetic position, is poorly understood. Our discovery of *U. rotundata* provides insights into this early evolution, namely the process of differentiation between the Sepiida and Sepiolida. Following the suggestion, we changed this part and focused on the early evolution of both coleoid groups in the revised manuscript (P. 3, L. 52–59 in the Article File).

Reference;

Hewitt, R. A. & Jagt, J. W. M. Maastrichtian Ceratisepia and Mesozoic cuttlebone homeomorphs. *Acta Palaeontol. Pol.* 44, 305–326 (1999).

P. 3, L. 53–54: Haas (2003) suggested the enigmatic genus *Belosepiella* as an ancestral sepiolid. Fuchs (2023) accordingly placed this genus as a supposed sepiolid. I think it is worth at least mention this idea of a Cenozoic sepiolid.

We thank Reviewer 1 for the valuable comments. As mentioned in Fuchs (2023), the assignment of *Belosepiella* to the fossil Sepiolida is still controversial. We thus rephrased this part as “The Sepiolida on the other hand have no reliable fossil record” in the revised manuscript, citing Fuchs (2023) (P. 3, L. 56 in the Article File).

P. 4, Fig. 2: is it possible to provide a larger figure?

Following the suggestion, we enlarged Fig. 2 (P. 4 in the Article File).

P. 4, L. 75: I assume it was not your aim to discover a sepioid beak, but to identify the owner of the beaks.

Following the suggestion, we rewrote this paragraph to start with “Here we report a new sepioid species”, avoiding the wording that we aimed to discover a sepioid beak, in the revised manuscript (P. 4–5, L. 80–104 in the Article File).

P. 5, L. 89: if I understand you correct, you identified the new species; not the AI, right? "...the system speeds up the visualization and thus the identification of the specimen"

As Reviewer 1 pointed out, AI did not conduct taxonomic identification of the new species. However, the fossils of the previously unknown species described in this study were first

detected by the AI used here. To make it precise, we changed “The system is, thus, able to detect previously unknown” to “The combination of grinding tomography and zero-shot learning AI is thus able to detect any fossils embedded in rocks even if they are taxonomically unknown” in the revised manuscript (P. 5, L. 87–89 in the Article File).

P. 5, Fig. 3: is it possible to provide a larger figure?

Following the suggestion, we enlarged Fig. 3. (P. 6 in the Article File).

P. 6, Fig. 4: is it possible to provide a larger figure?

Following the suggestion, we enlarged Fig. 4. (P. 8 in the Article File).

P. 8, L. 155: I recommend to describe the general preservation. To me it seems like internal lamellas are incomplete in both specimens

Following the suggestion, we described the general preservation of NMNS_DS00254_0k5cvkE.stl as “It is well-preserved except for its left wing and the posterior end of the lateral walls” in the revised manuscript (P. 10, L. 185–186, in the Article File). Similarly, we described that of NMNS_DS00285_ma2jp6.stl as “NMNS_DS00285_ma2jp6.stl is a small lower beak, well-preserved except for the wings and the posterior part of the lateral walls (Fig. 5h–l).” in the revised manuscript (P. 11, L. 213–214, in the Article File).

P. 11, L. 201: what exactly do you mean? coleoids or sepiolids?

We thank Reviewer 1 for the comment. This means coleoid here. We changed “cephalopods” to “coleoids” in the revised manuscript (P. 13, L. 240 in the Article File).

P. 11, L. 209: confusing terminology as internal shell is equivalent with gladius!
I would recommend to use "shell remains of coleoids" or "phragmocones, rostra, and gladiuses"

Following the suggestion, we changed “internal shells” to “phragmocones, rostra, and gladii” in the revised manuscript (P. 13, L. 250 in the Article File). For the plural form of “gladius”, we take the Latin form “gladii” rather than the English form “gladiuses”. This is consistent with the preceding word “rostra”, the Latin plural form of “rostrum”.

P. 11, L. 212: the paraphyletic taxon "Belemnnoidea" is outdated
better: Belemnitida represents a stem group of the Decabrachia (you Decapodiformes)

Following the suggestion, we changed “Belemnnoidea” to “Decabrachia” in the revised manuscript (P. 13, L. 252 in the Article File).

P. 11, L. 212: Belemnitida is represented by belemnites...?

We deleted “It is represented by belemnites and ” and connected the remaining part of this sentence to the preceding sentence in the revised manuscript (P. 13, L.252–253. in the Article File)

P. 11, L. 217: Octopoda or Octopodiformes?

We thank Reviewer 1 for the comment. In this part, we refer to “Octopoda”, not to “Octopodiformes”. We changed “Its fossils” to “The fossils of the Octopoda” in the revised manuscript (P. 13–14, L. 257–258 in the Article File).

P. 11, L. 217: s

Following the suggestion, we changed “a lentgh” to “lengths” in the revised manuscript (P. 14, L. 258 in the Article File)

P. 12, L. 219-220: which fossil record? I assume only the Western Interior fossil record? please name at least the genera; I don't know about 13 octopod species in the Western Interior

Following the suggestion, we noted all fossil genera of Octopoda from the Western Interior Seaway in the revised manuscript (P. 14, L. 260–261 in the Article File).

The list of 5 genera and 13 species in the original manuscript is as follows: *Actinosepia (canadensis, cf. canadensis, landmani, mapesi*: Larson 2010), *Muensterella (jillae*: Fuchs et al. 2020), *Enchoteuthis (melanae, cobbani, cf. cobbani, sp., sp., sp.*: Fuchs et al. 2020), *Niobrarateuthis (bonneri*: Fuchs et al. 2020), *Tusoteuthis (longa*: Fuchs et al. 2020). In the revised manuscript, we changed “13 species” to “8 species” by omitting taxa with open nomenclature (cf. or sp.) (P. 14, L. 260 in the Article File).

We also added the list of the belemnite genera in the same way as that for octopods in the revised manuscript (P. 13, L. 255–256 in the Article File).

References:

Larson, N. Fossil coleoids from the Late Cretaceous (Campanian & Maastrichtian) of the Western Interior. *Ferrantia* **59**, 78–113 (2010).

Fuchs, D. et al. The Muensterelloidea: phylogeny and character evolution of Mesozoic stem octopods. *Pap. Palaeontol.* **6**, 31–92 (2020).

Fuchs, D. Systematic descriptions: Octobranchia. *Treatise Online* **138**, 1–52 (2020).

P. 12, L. 223-224: please note that Belemnitida are nowadays dealt with as a member of the Decabrachia!

-> "...first record of crown decabrachians..."

Following the suggestion, we changed “the superorder Decapodiformes” to “non-belemnite decabrachians” in the revised manuscript (P. 14, L. 271 in the Article File). This change made this part consistent with the updated systematic assignment of the Belemnitida.

P. 12, L. 236-237: Sanchez et al. ignored fossils tentatively placed as sepiolids! see my annotations before

As Reviewer 1 pointed out, Sanchez et al. (2021) provided no explanation about the fossils tentatively assigned to sepiolids in Haas (2003). These fossils are, however, not a reliable fossil record for sepiolids. In the revised manuscript, we cited Fuchs (2023), who interpreted these fossils as controversial sepiolids (P. 15, L. 287 in the Article File).

References:

Fuchs, D. Systematic descriptions: Decabrachia. *Treatise Online* 166–177 (2023).

doi:10.17161/to.vi.21456.

P. 13, L. 260-262: this statement is - in my opinion - daring

We thank Reviewer 1 for the comment. Our findings cannot reconstruct the precise topology of the phylogenetic tree around sepioids. However, these findings can still show that the two sepioid orders are closely related, as Reviewer 1 mentioned in the general comments (P. 3 in this Point-by-Point Response File). We thus did not change this part in the revised manuscript (P. 16, L. 310–312 in the Article File).

Comments from Reviewer 2 (Remarks to the Author)

Overview.

This paper describes a new genus and species of sepioid cephalopod from the Cretaceous. This effort is a novel use of zero-shot learning (ZSL) on fossil data. This new genus and species sheds light upon the deep time history of this group and aids in understanding the timing of divergence between the Sepiida and Sepiolida, especially for groups with limited fossil records.

We thank Reviewer 2 for these positive comments. In the revised manuscript, we made it easier to understand the significance of this study, related to ZSL and the sepioid evolution.

Yasu Iba

Major Comments.

The paper provides minimal background on ZSL and provides minimal context for its use here. To engage the paleo community, it would help to discuss the advantages and drawbacks of this method, especially compared to other machine learning approaches.

Following the suggestion, we added a sentence to provide the background on ZSL for the paleo community in the revised manuscript as follows: “Zero-shot learning AI enables the detection of any objects in images regardless of whether they are already known or unknown, without requiring an additional training process^{29,30}” (P. 5, L. 85–87 in the Article File).

Was the approach attempted on a CT scan of the block before taking the serial sectioning approach?

We did not attempt a CT scan for the samples used in this study. The CT scan cannot visualize the internal structures of these samples sufficiently, due to the low density contrast between fossils and matrix (see Fig. 3 in Iba et al. 2025).

References;

Iba, Y. et al. Nature visible only digitally. *Patterns* (2025)
<https://doi.org/10.1016/j.patter.2025.101210>

Further, the discussion passage L241-249 might be misleading. As far as I can tell, the deep learning approach in this paper is for segmentation, not taxonomic identification. The latter would involve incorporating beaks for other taxa to evaluate proximity of the unknown object in the known, taxonomic embedding space. Thus, the usage appears to be more about object detection, not classification per se. I would hesitate to describe zero-shot as capable of discovering unknown fossils in a taxonomic sense without this added context, but rather as useful for discovering objects that *may* be new taxonomic discoveries.

As Reviewer 2 pointed out, “discovery” means the detection of fossils, not taxonomic identification in this context. To make it precise, we changed the explanation that our ZSL methods can discover “any fossils embedded in rocks even if they are taxonomically unknown” in the revised manuscript (P. 5, L. 87–89 in the Article File).

It's good to have the specs of the computing system used to fit the model (L285+), but it would be better to specify the arguments used for the run. Was the text-prompted mode for automatic mode used for DEVA? Were any non-standard arguments specified for the prompt?

Following the suggestion, we added the script used for our analysis and clarified the DEVA settings in the revised manuscript (P. 17, L. 341–344 in the Article File).

How was quality control and cross-checking of the segmentation output handled? Visual inspection?

We visually checked all output mask data to confirm that segmentation was done without excesses or deficiencies. We added an explanation for this in the revised manuscript (P. 17, L. 344–345 in the Article File).

Were the DEVA segmentation masks for the beak manually adjusted in Amira? More details on model results, any code, model tweaks, etc. would be critical for reproducing the results and allowing future applications of this method.

As manual adjustments, we imported the mask data to Amira, selected the lower beak masks, and converted them to label data. Following the suggestion, we added the detailed description of these operations in the revised manuscript (P. 17, L. 345–348 in the Article File).

Fig. 5 shows many views of the beak for the new taxon, which is helpful for visualizing its morphology. Does the limited view of beak in Fig. 4 show the full differences in morphology among the taxa, or would additional views help here?

Following the suggestion, we added some additional views in the revised Fig. 4 (P. 8 in the Article File).

Line Comments.

L29: “in which an AI model has led to”

Following the suggestion, we added “has” between “model“ and “led” in the revised manuscript (P. 2, L. 29 in the Article File).

L57: switch “only” and “exceeded”

Following the suggestion, we switched “only” and “exceeded” in the revised manuscript (P. 3, L. 62 in the Article File). We also modified the sentence structure of this part in the revised manuscript to be clearer (P. 3, L. 61–62 in the Article File).

L83: delete “an”

Following the suggestion, we delete “an” between “allowed” and “accurate” in the revised manuscript (P. 5, L. 100 in the Article File).

L89: delete the commas around “thus.” Also, it may to better connect the steps here as: “The system is thus able to isolate cohesive objects in matrix that may fossilized remains of unknown taxa.”

Following the suggestion, we deleted the commas and added connecting steps as follows in the revised manuscript (P. 5, L. 87–89 in the Article File): “The combination of grinding tomography and zero-shot learning AI is thus able to detect any fossils embedded in rocks even if they are taxonomically unknown”.

We also received similar comments on this part from Reviewers 1 and 4. We thus changed the sentence as shown above, considering all the comments.

L90 & 91: delete “old” after both instances of Ma

Following the suggestion, we delete “old” in the revised manuscript (P. 4, L. 81 in the Article File)

L91-92: what was the previous oldest known sepioid? Where was it from?

Ceratisepia vanknippenbergi from the late Maastrichtian of the Netherlands (Hewitt and Jagt 1999) was the previous oldest known sepioids. We explained it in the revised manuscript (P.3, L. 52–56, and P. 14–15, L. 280–282 in the Article File).

Reference;

Hewitt, R. A. & Jagt, J. W. M. Maastrichtian *Ceratisepia* and Mesozoic cuttlebone homeomorphs. *Acta Palaeontol. Pol.* 44, 305–326 (1999).

L107: should this be “the inner edge of either bridge is connected...”?

Following the suggestion, we changed “the bridges” to “either bridge” in the revised manuscript (P. 6, L. 119 in the Article File)

L109: delete comma after Octopoda

Following the suggestion, we deleted the comma in the revised manuscript (P. 6, L. 121 in the Article File)

L144: delete “the” before “Timber Lake Member”

Following the suggestion, we delete “the” before “Timber Lake Member” in the revised manuscript (P. 9, L. 172 in the Article File)

L201: change “on” to “of”

Following the suggestion, we changed “on” to “of” in the revised manuscript (P. 13, L. 241 in the Article File).

L229-230: suggest: “thrives globally in the shallow areas of all oceans except along the coasts of North and South America”

Following the suggestion, we changed “thrive globally in the shallow areas of oceans except for North and South America” to “thrive globally in the shallow areas of oceans except along the

coasts of North and South America” in the revised manuscript (P. 14, L. 278–279 in the Article File).

L236: add “serve as” between “and” and “model”

Following the suggestion, we added “serve as” between “and” and “model” in the revised manuscript (P. 15, L. 286 in the Article File).

L256-257: I would reword as “even with the extensive morphological and molecular data available”

Following the suggestion, we changed “even with extensive morphological and molecular data being available” to “even with the extensive morphological and molecular data available” in the revised manuscript (P. 16, L. 306–307 in the Article File).

L263: Add an “and” between “Cretaceous” and “Sepiida”

Following the suggestion, we add an “and” between “Cretaceous,” and “the Sepiida” in the revised manuscript (P. 16, L. 313 in the Article File).

L272 & 275: Does Zone need to be capitalized?

Both capitalized and not capitalized ones are accepted in Communications Biology, we kept “Zone” capitalized in the revised manuscript (P. 16, L. 322 and 325 in the Article File).

L282: Add an “and” between “pixels” and “the”

Following the suggestion, we added an “and” between “pixels” and “the” in the revised manuscript (P. 17, L. 332 in the Article File).

L322: Remove commas around “therefore”

Following the suggestion, we removed commas in the revised manuscript (P. 19, L. 380 in the Article File).

Comments from Reviewer 3 (Remarks to the Author)

I co-reviewed this manuscript with one of the reviewers who provided the listed reports. This is part of the Communications Biology initiative to facilitate training in peer review and to provide appropriate recognition for Early Career Researchers who co-review manuscripts.

We thank Reviewer 3 for carefully checking our manuscript. We highly appreciate this far-seeing and ambitious initiative from *Communications Biology*.

Yasu Iba

Comments from Reviewer 4 (Remarks to the Author)

Dear authors,

It is a great idea to apply grinding tomography to find fragile fossil structures such as non-mineralized cephalopod beaks. In that sense, I very much welcome your manuscript!

Additionally, the use of AI is novel.

Further, new occurrences of such cephalopod beaks are important because they are rare and fill big gaps in our knowledge.

We thank Reviewer 4 for all the positive comments and for carefully checking our manuscript.

The problem with coleoid beaks is that the morphology differs rather subtly between taxa, making their identification difficult.

I made several annotations in the pdf and highlight my main points of criticism here:

1. I am not convinced that this is a sepiid beak. For a sepiolid, it is on the giant side and to me, it does not look intermediate between the two sepiid clades but rather like lying phylogenetically outside.

This shows a fundamental problem: How do you place the new species in systematics? You could use phylogenetic (Bayesian, ma. likelihood) or morphometric methods but there is neither mentioned in the main text, but it should be. Could it be that it is a form in the stem of the two groups?

This would make most sense to me, presuming your assignment to that clade holds true (I am not sure about this, though).

We thank Reviewer 4 for the valuable comment. We conducted the morphological disparity analysis of the lower beak for *Uluciala* and 165 modern cephalopod species (P. 14, L. 275–276 in the Article File). These taxa cover all extant orders and >90% of all modern cephalopod families. The details of the analysis and its results are shown in the Supplementary Discussion and Supplementary Figs. 1 and 2 (P. 2–4 in Supplementary Information File). In the results, *Uluciala* is closer to the Sepiida and Sepiolida than other groups, and is placed at an intermediate position between the two orders. These results support our systematic assignment.

We also added non-sepioid coleoid groups to the revised Fig. 4 for comparison of the lower beak morphology (P. 8 in the Article File).

Sizes of modern Sepioids: We are doubtful whether the body size of fossil taxa can help to assign these forms to the Sepiolida, because it is yet unknown when they diminished in size. It is possible that ancestral Sepiolida include species with a large body size. We thus did not change the manuscript parts related to this topic.

2. In places it sounds like you were the first using grinding tomography, which is not true. My work-group has published a series of paper employing this method to cephalopod conchs (Tajika et al. 2015, 2018, Naglik et al. 2015, 2016) and there are older papers on bivalves (Götz 2003) and even older ones by Stensiö or so, because before CT became standard, vertebrate, coral and brachiopod workers used this technology, although by hand. I think this should be properly credited.

Following the suggestion, we cited some pioneering studies (Götz, 2007; Naglik et al. 2015; Mehra et al. 2020) that applied grinding tomography to fossils in the revised manuscript (P. 17, L. 331 in the Article File).

In the past, grinding tomography has focused on the study of internal structures of target fossils that were already exposed from their host rock. In contrast, we used grinding tomography to discover any fossils embedded in whole rocks. Our approach is thus based on a concept deviating from previous studies. In the revised manuscript, we clarified the advancements of our approach, which combined grinding tomography and zero-shot learning AI to discover fossils of previously unknown taxa (P. 5, L. 84–89 in the Article File).

References;

- Götz, S. Inside rudist ecosystems: growth, reproduction, and population dynamics. in *Cretaceous Rudists and Carbonate Platforms: Environmental Feedback* (ed. Scott, R. W.) SEPM Special Publication vol. 87 (Society for Sedimentary Geology, 2007).
- Naglik, C., Monnet, C., Götz, S., Kolb, C., De Baets, K. and Klug, C. Growth trajectories in chamber and septum volumes in major subclades of Paleozoic ammonoids. *Lethaia* **48** 29–46 (2015).
- Mehra, A., Watters, W. A., Grotzinger, J. P. & Maloof, A. C. Three-dimensional reconstructions of the putative metazoan Namapoikia show that it was a microbial construction. *Proc. Natl. Acad. Sci.* **117**, 19760–19766 (2020).

3. The figure explaining the terminology is hard to understand.

Following the suggestion, we added magnified views with more detailed explanations in the revised Fig. 2, making it easier to understand (P. 4 in the Article File).

4. Conclusions are provided before the discussion in lines 91–93. I would remove this and limit such interpretations to after the description and comparison.

We thank Reviewer 4 for the comment. The guideline of *Communications Biology* requires that the final paragraph of introduction should be a brief summary of the major results and conclusions. Thus, we kept these brief conclusions in the introduction section, following this guideline (P. 4–5, L. 80–104, in the Article File).

Reference:

Style and formatting guide. *Communications biology*
url: <https://www.nature.com/documents/commsj-life-style-formatting-guide-accept.pdf>

5. The comparison is limited (at least visually in Fig. 4) to sepiids.

I find that the beaks also resemble Ommastrephes, but maybe I have missed some detail. For the reader, it would be great if representative lower jaws of other decabrachian clades were included in Fig. 4 and not just their lateral aspects. Maybe you are right and I am wrong, but I am just not convinced about the systematic assignment.

Following the suggestion, we added non-sepioid coleoid groups to the revised Fig. 4 (P. 8 in the Article File).

Uluciala can be clearly distinguished from *Ommastrephes*, which belongs to the Oegopsida, by the absence of shoulders and its remarkably large rostral hook. The comparison with the Oegopsida is noted in the order-level remarks in the Systematic paleontology section (P. 6, L. 122–123, in the Article File). The morphological disparity analysis further clarified that *Uluciala* is much closer to sepioids than to *Ommastrephes* and other oegopsids (P. 2–4, in the Supplementary Information File).

6. For sepiolids, the beaks would be on the giant side, how do you explain this?

We are doubtful whether fossil taxa can be assigned to the Sepiolida by body sizes since it is yet unknown when they diminished in size. It is possible that ancestral Sepiolida include species with a large body size. We thus did not change the manuscript parts related to this topic.

7. Are there any remains of upper beaks or gladius remains? In such early forms, I would expect that they still had some internal sclerotized support with fossilization potential comparable to beaks. This should be shortly discussed in either case.

We discovered some upper beaks from the concretions used in this study. We did not, however, mention these fossils since the upper beaks cannot be used taxonomically at the order level (P. 13, L. 238–240, in the Article File).

Fossils of the internal sclerotized support, namely gladius or phragmocone, were not discovered in this study. This absence is possibly caused by the composition of the gladius, which is much easier to decompose than that of beaks (Tan et al. 2015). We added a more detailed review of the gladius or phragmocone fossils from the Western Interior Seaway in the revised manuscript (P. 13–14, L. 249–263 in the Article File).

Reference;

Tan, Y. et al. Infiltration of chitin by protein coacervates defines the squid beak mechanical gradient. *Nat. Chem. Biol.* **11**, 488–495 (2015).

The other comments are of secondary importance but should still be looked at.

Best regards,
Christian Klug

We thank Reviewer 4 for all the positive comments and insightful feedback on the work. We added point-by-point responses for the comments in the attached PDF on the following pages.

Yasu Iba

Comments from Reviewer 4 (Attached PDF)

P. 1, L. 23: I'd prefer internally shelled, but Neil Landman as native speaker knows best.

Following the suggestion, we changed “internal-shelled” to “internally shelled” in the revised manuscript (P. 2, L. 23 and P. 3, L. 38 in the Article File).

P. 2, L. 36-37 :

Same as for “internally shelled” in P. 2, L. 23 in the revised manuscript, we changed “externall-shelled” to “externally shelled” in the revised manuscript (P. 2, L. 36–37 in the Article File).

P. 2, L. 37: ammonoids (otherwise, these are only Mesozoic)

Following the suggestion, we changed “ammonites” to “ammonoids” in the revised manuscript (P. 2, L. 37 in the Article File).

P. 2, L. 37: systematically, this is probably not what you want to say, because there are many groups that are technically not nautiloids and still have external conchs:

Orthoceratoidea

Multiceratoidea

Endoceratoidea

Following the suggestion, we changed “namely the ammonites and nautiloids” to “such as ammonoids and nautiloids” in the revised manuscript (P. 2, L. 37 in the Article File).

P. 2, L. 38: I would be less specific, because Ward will likely describe more. Replace by modern or Recent

Following the suggestion, we changed “nine species of external-shelled nautiloids” to “modern nautiloids” in the revised manuscript (P. 2, L. 38 in the Article File).

P. 2, L. 40: correct: Superorder Decabrachia Haeckel, 1866

Reference: Hoffmann, R., Howarth, M. K., Fuchs, D., Klug, C. & Korn, D. (2022): The higher taxonomic nomenclature of Devonian to Cretaceous ammonoids and Jurassic to Cretaceous ammonites including their authorship and publication. – Neues Jahrbuch für Geologie und Paläontologie, 305/2: 1-11

Following the suggestion, we followed Hoffman et al. 2022 and changed “Decapodiformes” to “Decabrachia” in the revised manuscript (e.g., P. 2, L. 39 in the Article File).

P. 3, Fig. 1: Decabrachia

Following the suggestion, we changed “Decapodiformes” to “Decabrachia” in the revised manuscript (P. 3, L. 48 in the Article File).

P. 3, L. 53-54: It would be worth mentioning, why: Small size, reduced gladius I would also cite this here: Clements, T., Colleary, C., De Baets, K., & Vinther, J. (2016). Buoyancy mechanisms limit reservation of coleoid cephalopod soft tissues in Mesozoic lagerstätten. Palaeontology, 60, 1–14.

Following the suggestion, we mentioned why they lack fossil records, citing Clements et al. 2016, in the revised manuscript (P. 3, L. 56–58 in the Article File).

P. 3, L. 60: I would also cite one of these papers: Roscian, M., Herrel, A., Cornette, R., Delapré, A., Cherel, Y., and Rouget, I. (2021). Underwater photogrammetry for close-range 3D imaging of dry-sensitive objects: The case study of cephalopod beaks. *Ecol. Evol.* 11 (12), 7730–7742. doi:10.1002/ece3.7607

Roscian, M., Herrel, A., Zaharias, P., Cornette, R., Fernandez, V., Kruta, I., et al. (2022). Every hooked beak is maintained by a prey: Ecological signal in cephalopod beak shape. *Funct. Ecol.* 36, 2015–2028. doi:10.1111/1365-2435.14098

Following the suggestion, we added these citations in the revised manuscript (P. 3, L. 64 and P. 13, L. 238 in the Article File).

P. 4, L. 74-75: This sounds strange. You don't go to the field and say: I discover sepioid beaks. Please change the wording.

Following the suggestion, we rewrote this paragraph to start with “Here we report a new sepioid species”, avoiding the wording that we aimed to discover a sepioid beak, in the revised manuscript (P. 4–5, L. 80–104 in the Article File).

P. 5, L. 82: this term needs to be explained at the first occasion it is mentioned or at least very early on.

Following the suggestion, we added the following explanation of Zero-shot learning immediately after it is first mentioned in the revised manuscript: “Zero-shot learning enables the detection of any objects in images regardless of whether they are already known or unknown, without requiring an additional training process” (P. 5, L. 85–87 in the Article File).”

P. 5, L. 91: This is a conclusion BEFORE the discussion!

P. 5, L. 91: This needs to be organized differently in my opinion: Describe the material, compare, THEN conclude.

We thank Reviewer 4 for the comment. The guideline of *Communications Biology* requires that the final paragraph of introduction should be a brief summary of the major results and conclusions. Thus, we kept this brief conclusion in the introduction section, following this guideline (P. 5, L. 100–104, in the Article File).

Reference:

Style and formatting guide. *Communications biology*

url: <https://www.nature.com/documents/commsj-life-style-formatting-guide-accept.pdf>

P. 6, L. 106: In your Figure 2, it is not at all clear what you mean by bridges. This needs a Figure, where this is really clear (larger and distinct from other structures).

Following the suggestion, we made it clearer by adding magnified views showing the parts around bridges in the revised Fig. 2 (P. 4, L. 70–78, in the Article File).

P. 6, L. 106: It would be nice to see all these groups represented in Fig.4.

Following the suggestion, we added the lower beak morphology of all these groups (Octopoda, Vampyromorpha, Spirulida, Oegopsida, Myopsida, and Belemnitida) in the revised Fig. 4 (P. 8 in the Article File).

P. 6, L. 113: in what sense? In bemenites, it is much longer, so do you mean broad or strong?

Here we described that “the hook is large” in terms of width. We explained the relative width of the hook compared to the distance between the ventral end of the lateral wall on both sides in the revised manuscript (P. 10, L. 191–192 and P. 11, L. 222–223 in the Article File). This ratio is ~0.30 in the Belemnitida (Klug et al. 2010, Fig. 5, 6c), while it is >0.40 in *U. rotundata*. These values suggest that the lower beak of *U. rotundata* has a larger hook than that of the Belemnitida.

Reference;

Klug, C., Schweigert, G., Fuchs, D. & Dietl, G. First record of a belemnite preserved with beaks, arms and ink sac from the Nusplingen Lithographic Limestone (Kimmeridgian, SW Germany). *Lethaia* **43**, 445–456 (2010).

P. 6, L. 113: I am not a great fan of abbreviations and in this case, I think it is misleading. You are talking about LOWER jaws, so this edge is facing more or less anteriorly, unless the animal is swimming head down.

Following the suggestion, we rephrased the sentences in Systematic paleontology without using abbreviations in the revised manuscript (e.g., P. 7, L. 131 and 133–134 in the Article File). The cephalopod lower beak is traditionally described in a position upside-down to the body (e.g. Clarke 1986; Young et al. 2007). We thus followed that rule in this study.

P. 6, L. 114: I wouldn't use it without 'on the one hand'. Maybe replace by: 'By contrast' or similar.

Following the suggestion, we changed “On the one hand” to “By contrast” in the revised manuscript (P. 7, L. 133 in the Article File)

P. 6, L. 116: This is also similar in Ommastrephes (Roscián 2021, Fig. 3d). I am not convinced.

Uluciala can be clearly distinguished from *Ommastrephes*, which belongs to the Oegopsida, by the absence of shoulders and its remarkably large rostral hook. The comparison with the Oegopsida is summarized in the order-level remarks in Systematic paleontology (P. 6, L. 122–123, in the Article File). We also added non-sepioid coleoid groups, including the Oegopsida, to the revised Fig. 4 (P. 8 in the Article File). The morphological disparity analysis further clarified that *Uluciala* is much closer to sepioids than to *Ommastrephes* (P. 2–4, in the Supplementary Information File).

P. 7, L. 137-138: Would it make sense to also deposit 3D-prints?

Since our digital 3D models will be available from an open-access depository, they can be printed using 3D printers in high resolution, which is easier than visiting museums to see 3D prints. In the revised manuscript, we added the following sentence to Systematic paleontology

section: “These 3D models are also openly accessible from Figshare (doi: 10.6084/m9.figshare.28119998)³⁸ and can be freely viewed through various types of software or 3D printing.” (P. 9, L. 162–166 in the Article File). We also included the following sentence to Methods section: “As these data are given in universal formats, they can be freely viewed through various types of software or 3D printing.” (P. 19, L. 393–394 in the Article File).

P. 8, L. 157: as stated above, if abbreviations are not fully established, they make texts awful to read. I recommend to write them out the first time you use them HERE in the description to help the reader understand WHAT you are writing about.

Following the suggestion, we modified the sentences in Systematic paleontology and Fig. 2 without using abbreviations in the revised manuscript (P. 10, L. 188 in the Article File).

P. 8, L. 158: This sounds gigantic for a sepiolid.

We are doubtful whether fossil taxa can be assigned to the Sepiolida by body sizes since it is yet unknown when they diminished in size. It is possible that ancestral Sepiolida include species with a large body size. We thus did not change this part in the revised manuscript (P. 10, L. 188 in the Article File).

P. 9, L. 179: Are there any upper jaws or gladius remains?

We discovered some upper beaks from the concretions analyzed in this study. We did, however, not mention these fossils since the upper beaks cannot be used taxonomically at the order level (P. 13, L. 238–240, in the Article File).

Fossils of gladius remains were not discovered in this study. We added a more detailed review of the gladius or phragmocone fossils from the Western Interior Seaway in the revised manuscript (P. 13–14, L. 249–263 in the Article File).

P. 10, Fig. 5: that notch is conspicuous. Caused by wear?

As Reviewer 4 noted, this notch might have been formed by wear. We did not, however, discuss the cause because the critical evidence is missing to distinguish whether this notch is an original form or caused by wear.

P. 10, L. 193: at a mantle length of up to 8 cm, modern sepiolids probably have smaller jaws.

Modern Sepiolida are usually smaller than 8 cm in mantle length, but it is yet unknown when they diminished in size. It is possible that ancestral Sepiolida include species with a large body size. We thus did not change the manuscript about this matter.

P. 11, L. 199: that is a bit overgeneralized, I find. Some groups do have characteristic upper beaks. I would tone this down a bit.

Following the suggestion, we changed “Only the lower beak is taxonomically relevant” to “Only the lower beak was used here for the taxonomic identification” in the revised manuscript (P. 13, L. 238–239 in the Article File).

P. 11. L. 201: cephalopods without or only weakly sclerotized internal skeleton

Following the suggestion, we changed “shell-less cephalopod” to “coleoids without or only weakly sclerotized internal skeleton” in the revised manuscript (P. 13, L. 240–241 in the Article File).

P. 11. L. 205: an important source of data contributing...

Following the suggestion, we changed “an alternative approach” to “an important source of data contributing” in the revised manuscript (P. 13, L. 245 in the Article File).

P. 11. L. 211:

Following the suggestion above, we changed “internal-shelled” to “internally shelled” in the revised manuscript (P. 13, L. 252 in the Article File).

P. 11. L. 212: Hoffmann et al. (2022) show a different systematic structure.

Following the suggestion, we changed “Belemnnoidea” to “Decabrachia” in the revised manuscript, according to Hoffmann et al. (2022) (P. 13, L. 252 in the Article File).

P. 12. L. 227: well, it is morphologically different but I would not say intermediate.

Following the suggestion, we changed “is an intermediate taxon” to “had an intermediate morphology” in the revised manuscript (P. 14, L. 274 in the Article File). We also added a sentence in the revised manuscript: “This result indicates that *U. rotundata* is likely a taxon showing the process of differentiation between the two orders.” (P. 14, L. 276–277 in the Article File).

P. 12. L. 229: IMO Sepiida is plural. Unless you write 'the clade Sepiida...'

Following the suggestion, we changed “thrives” to “thrive” in the revised manuscript (P. 14, L. 278 in the Article File). We also changed other sentences, of which the subject is an order name, to plural in the revised manuscript (e.g., P. 14, L. 277 in the Article File).

P. 12. L. 230: Currently, the

Following the suggestion, we changed “The oldest...” to “Currently, the oldest...” in the revised manuscript (P. 14, L. 280 in the Article File).

P. 12. L. 231: delete

Following the suggestion, we deleted “southern” in the revised manuscript (P. 15, L. 282 in the Article File).

P. 13. L. 243: is this relevant here?

This use of AI is relevant here for the following reasons. In this sentence, we introduce the general uses of AI in paleontology. Therefore, we have to list taxonomic identification, which

covers most of the AI uses in paleontology, along with image segmentation. Also, we would like to show here that our methods have a contrasting purpose to the AI models for taxonomic identification. This study used AI methods to discover fossils of unknown taxonomic affiliation, but past AI methods, developed for taxonomic identification, are focused only on taxa already described. Reviewer 2 also made a comment on this part, agreeing that this background information will provide a better understanding for the paleontological community. Thus, we kept this sentence in the revised manuscript (P. 15, L. 293 in the Article File).

P. 14. L. 283-284: There are much earlier papers, please cite them!*--

For cephalopods:

Naglik, C., Monnet, C., Götz, S., Kolb, C., De Baets, K. & Klug, C. (2015): Growth trajectories in chamber and septum volumes in major subclades of Paleozoic ammonoids. – *Lethaia*, 48: 29-46

Tajika, A., Naglik, C., Morimoto, N., Pascual-Cebrian, E., Hennhöfer, D. K. & Klug, C. (2015): Empirical 3D-model of the conch of the Middle Jurassic ammonite microconch *Normannites*, its buoyancy, the physical effects of its mature modifications and speculations on their function. – *Historical Biology*, 27(2): 181-191

In general:

Götz S, Stinnesbeck W. 2003. Reproductive cycles, larval mortality and population dynamics of a Late Cretaceous hippuritid association: a new approach to the biology of rudists based on quantitative three dimensional analysis. *Terra Nova* 15:392–397 DOI 10.1046/j.1365-3121.2003.00515.x.

Following the suggestion, we cited some pioneering studies (Götz, 2007; Naglik et al. 2015; Mehra et al. 2020) that applied grinding tomography to fossils in the revised manuscript (P. 17, L. 331 in the Article File).

The previous uses of the grinding tomography have focused on the internal structures of target fossils either already exposed on or fully separated from rocks. In contrast, we used the grinding tomography for discovering fossils embedded in whole rocks. Our approach is thus based on different concepts from previous studies. In the revised manuscript, we clarified the advancements of our approach, which combined grinding tomography and zero-shot learning AI to discover fossils of previously unknown taxa (P. 5, L. 84–89 in the Article File).

References;

Götz, S. Inside rudist ecosystems: growth, reproduction, and population dynamics. in *Cretaceous Rudists and Carbonate Platforms: Environmental Feedback* (ed. Scott, R. W.) SEPM Special Publication vol. 87 (Society for Sedimentary Geology, 2007).

Naglik, C., Monnet, C., Götz, S., Kolb, C., De Baets, K. and Klug, C. Growth trajectories in chamber and septum volumes in major subclades of Paleozoic ammonoids. *Lethaia* **48** 29-46 (2015)

Mehra, A., Watters, W. A., Grotzinger, J. P. & Maloof, A. C. Three-dimensional reconstructions of the putative metazoan *Namapoikia* show that it was a microbial construction. *Proc. Natl. Acad. Sci.* **117**, 19760–19766 (2020).

P. 16. L. 324-329: I recommend to store 3D-prints somewhere.

Since our digital 3D models will be available from an open-access depository, they can be printed using 3D printers, which is easier than visiting museums to see 3D prints. In the revised manuscript, we included the following sentence to Systematic paleontology section: “These 3D

models are also openly accessible from Figshare (doi: 10.6084/m9.figshare.28119998)³⁸ and can be freely viewed through various types of software or 3D printing.” (P. 9, L. 162–166 in the Article File). We also included the following sentence to Methods section: “As these data are given in universal formats, they can be freely viewed through various types of software or 3D printing by a personal printer.” (P. 19, L. 393–394 in the Article File).

Point-by-Point Response to the Reviewers

COMMSBIO-25-4252B

We sincerely appreciate the reviewers' valuable and insightful comments on our manuscript. We have accepted all the major suggestions and comments made by the reviewers, thereby improving the revised manuscript.

In the point-by-point response below (P. 2–5), the comments from the reviewers are underlined, our response is given in **blue color**. The final adjustments are all integrated in the Article File and are highlighted in **red color**.

The comments and responses are ordered as follows: Comments from Reviewer 1 (P. 2), Comments from Reviewer 4 (P. 3), Comments from Reviewer 5 (P.4–5).

Comments from Reviewer 1

Dear authors, you adopted the main points the referees recommended in their first review. This improved the ms significantly.

Moreover, you supported your systematic interpretation by additional statistical analyses. Although I agree with your decision of an open nomenclature and although I still think your evolutionary/phylogenetic conclusions are immature, I recommend to accept the present ms as it is.

Dirk Fuchs

We thank Reviewer 1 for approving our additional statistical analyses and for accepting our manuscript. Considering your comment about “immature conclusions” we clearly stated that our evolutionary interpretation is supported by morphological analyses in the revised manuscript (P. 14, L. 280–282, in the Article File).

Yasu Iba

Comments from Reviewer 4

Dear Yasu and co-authors,

I am happy with the ms now!

I look forward to see it published!

Best wishes,

Christian

We thank Reviewer 4 for accepting our manuscript.

Yasu Iba

Comments from Reviewer 5

This article uses a zero-shot learning model to process and segment fossil images, separating the fossils from the surrounding rocks and constructing a three-dimensional model of the fossils. Eventually, an important new species was discovered. This article combines AI methods with paleontology, which is a typical interdisciplinary achievement and is worthy of publication.

We thank Reviewer 5 for the all positive comments and for carefully checking our manuscript. We added point-by-point responses for the comments in the attached PDF on the following pages.

Major issue:

An artificial intelligence model using zero-shot learning discovered a new type of fossil species. This statement is ambiguous, including the title of the article. In fact, the authors merely used the zero-shot learning method to process fossil images and separate them from the surrounding rocks. The zero-shot learning model in this article mainly played the roles of image processing and segmentation, rather than directly participating in the discovery and classification of new species. The discovery of the new species was not based on this model, but was rather determined by human analysis of the fossil's characteristics.

Following the suggestion from Reviewer 5, we clarified that the AI model did not conduct the taxonomic identification in the revised manuscript. We explained that the identification of the new fossil taxon was based on our systematic classification and morphological analyses in the revised main text and Methods (P. 18, L. 366 and P. 20, L. 406, in the Article File). We also clearly indicated that the AI model still played an essential role in the discovery of the new taxon by isolating its fossils (P. 15, L. 304–307, in the Article File).

In paleontology, technologies for isolating fossils from rocks are a key to discovering previously unknown taxa, a process that is even more critical than the subsequent taxonomic identification. However, there have been no technological innovations that enable the indiscriminate isolation of fossils from rocks, while many computational techniques have been developed for taxonomic identification (e.g., morphometry). Although it is the biggest problem in fossil-based studies, most paleontologists have overlooked it until Ikegami et al. (2025) pointed it out.

Recently, Ikegami et al. (2025) developed the Digital fossil-mining method, which is clearly an advancement for solving this problem. The Digital fossil-mining method first converts whole rocks into image datasets by grinding tomography (P. 4–5, L. 84–86, in the Article File). Fossils in the datasets are then isolated from rocks through segmentation and are visualized as 3D models (P. 5, L. 86–87, in the Article File). However, even with this method, fossils are still isolated through manual segmentation, biased by the researchers' recognition and knowledge. Previous AI-based methods developed in paleontology also did not solve this problem since they are based on training data biased by manual segmentation (e.g., Knutsen et al. 2024).

In this study, we successfully isolated all fossils from rocks without human-induced biases by incorporating a zero-shot learning AI model to the Digital fossil-mining method. Zero-shot learning AI has a learning algorithm different from conventional AI methods, thereby it can detect any object even if they are not included in the training data. Our methods are thus able to facilitate discoveries of unexpected fossils, including previously unknown taxa.

We agree that the usage of the word “discovery” in our original manuscript could lead to misunderstandings. We thus changed the manuscripts as follows:

1. We added “Digital fossil-mining with” in the title (P. 1, L. 1–2, in the Article File).
2. We changed “This is the first study in which an AI model has led to the discovery of a new fossil taxon” to “This discovery was facilitated by a new approach in paleontology, the Digital fossil-mining method incorporating a zero-shot learning AI model.” (P. 2, L. 27–29, in the Article File).
3. We added the explanations of the Digital fossil-mining method, and the AI model for segmentation to the Introduction section (P. 4–5, L. 82–88, in the Article File).
4. We deleted “discovery” in the caption of Fig. 3, by changing “The process of discovering unknown fossil taxa by a zero-shot learning AI” to “The process of the Digital fossil-mining method that incorporates a zero-shot learning AI” (P. 6, L. 109, in the Article File).
5. We changed our review of previous AI applications in paleontology to focus specifically on segmentation in the Discussion section (P. 15, L. 298–304, in the Article File).
6. We changed “applicable to any material” to “able to excavate any fossils from tomographic datasets” (P. 15, L. 304–305, in the Article File).
7. We changed “Grinding tomography and automatic segmentation by DEVA” to “The Digital fossil-mining method combined with a zero-shot learning AI” (P. 17, L. 336, in the Article File).

Our changes clarified that the AI model has conducted a key process in the discovery of the new taxa described here, namely isolating its fossils. We thus kept the term “discovery” in the appropriate contexts of the revised manuscript. It also makes the manuscript easier to understand for both specialists and general readers.

Alternatively, the term “discovery” could be replaced with “visualization”; however, this modification would obscure the essence of this study.

References;

Ikegami, S., Takeda, Y., Mutterlose, J., and Iba, Y. Origin and radiation of squids revealed by digital fossil-mining. *Science* **388**, 1406–1409 (2025).

Knutsen, E. M. & Kononov, D. A. Accelerating segmentation of fossil CT scans through Deep Learning. *Sci. Rep.* **14**, 20943 (2024).

Minor issue

Lines 292-293 AI models have also been adopted in obtaining morphological characters from fossils, e.g. Liu et al 2024 Nature Ecology & Evolution

According to the above comments from Reviewer 5, we changed our review of previous AI applications in paleontology to focus specifically on segmentation (P. 15, L. 298–304, in the Article File). We therefore do not mention Liu et al. (2024).

Line 330 “converted into ~3,000 cross-sectional images”, here needs a precise number of images

Following the suggestion, we changed “~3,000” to “2,970 and 2,638” in the revised manuscript (P. 17, L. 338, in the Article File).

Yasu Iba